# Monocyte depletion enhances neutrophil influx and proneural to mesenchymal transition in glioblastoma

Zhihong Chen [1,2,3] ✉, Nishant Soni[4,14], Gonzalo Pinero[1,14], Bruno Giotti[4], Devon J. Eddins [2,5], Katherine E. Lindblad[1,6,7,8], James L. Ross[9], Montserrat Puigdelloses Vallcorba[1], Tanvi Joshi[1], Angelo Angione[1], Wes Thomason[1], Aislinn Keane[1], Nadejda M. Tsankova[10], David H. Gutmann [11], Sergio A. Lira [12], Amaia Lujambio [1,6,7,8], Eliver E. B. Ghosn [2,5], Alexander M. Tsankov[4] & Dolores Hambardzumyan [1,2,3,13] ✉

Myeloid cells comprise the majority of immune cells in tumors, contributing to tumor growth and therapeutic resistance. Incomplete understanding of myeloid cells response to tumor driver mutation and therapeutic intervention impedes effective therapeutic design. Here, by leveraging CRISPR/Cas9-based genome editing, we generate a mouse model that is deficient of all monocyte chemoattractant proteins. Using this strain, we effectively abolish monocyte infiltration in genetically engineered murine models of de novo glioblastoma (GBM) and hepatocellular carcinoma (HCC), which show differential enrichment patterns for monocytes and neutrophils. Eliminating monocyte chemoattraction in monocyte enriched *PDGFB*-driven GBM invokes a compensatory neutrophil influx, while having no effect on *Nf1*-silenced GBM model. Single-cell RNA sequencing reveals that intratumoral neutrophils promote proneural-to-mesenchymal transition and increase hypoxia in *PDGFB*-driven GBM. We further demonstrate neutrophil-derived TNF-a directly drives mesenchymal transition in *PDGFB*-driven primary GBM cells. Genetic or pharmacological inhibiting neutrophils in HCC or monocyte-deficient *PDGFB*-driven and *Nf1*-silenced GBM models extend the survival of tumor-bearing mice. Our findings demonstrate tumor-type and genotype dependent infiltration and function of monocytes and neutrophils and highlight the importance of targeting them simultaneously for cancer treatments.

The strong interdependence between neoplastic and non-neoplastic cells in the tumor microenvironment (TME) is a major determinant underlying cancer growth. The glioblastoma (GBM) TME is composed of a wide variety of non-neoplastic stromal cells, including vascular endothelial cells, various infiltrating and resident immune cells, and other glial cell types[1–3]. The predominant cell type in the GBM TME in both humans and murine tumor models are innate immune cells called tumor-associated myeloid cells (TAMs), which have been shown to promote tumor growth, invasion, and therapeutic resistance[2]. In GBM, TAMs are composed of mixed populations, the most abundant of which are of hematopoietic origin, including monocytes and monocyte-derived macrophages (MDM). Less abundant, although still a significant presence, are brain intrinsic microglia (Mg) and hematopoietic-derived neutrophils[4]. As such, it became appealing that

treatment aiming at obliterating myeloid cells could offer promising outcomes for GBM patients. However, despite extensive efforts over the past decade, macrophage-targeted therapies, either alone or in combination with standard-of-care therapy (SOC; radiation therapy [RT] and temozolomide [TMZ]), have largely failed in GBM clinical trials[5,6].

The Cancer Genome Atlas (TCGA) provides robust gene expression-based identification of GBM subtypes, including proneural (PN), mesenchymal (MES), and classical (CL) groups[7–10]. These subtypes are established based upon the dominant transcriptional patterns at the time and location of tumor resection, and are not mutually exclusive of each other, i.e., multiple subtypes can co-exist within a single tumor, both at the regional[11] and single-cell levels[12]. Aimed at defining a unified model of cellular and genetic diversity, one study found that malignant cells in GBM exist in four major plastic cellular states[13] that closely resemble distinct neural cell types, including: neural progenitor-like (NPC-like), oligodendrocyte progenitor-like (OPC-like), astrocyte-like (AC-like), and mesenchymal-like (MES-like) states[13]. Tumors with a MES-like state demonstrate striking similarities to the TCGA-MES subtype, where both are enriched with TAMs[10,14,15]. Using genetically engineered mouse models (GEMMs) that closely resemble human PN, CL, and MES subtypes, we previously showed that driver mutations define myeloid cell composition in tumors[4]. In contrast to *PDGFB*-overexpressing tumors (resembling human PN GBM) or *EGFRvIII*-expressing tumors (resembling human CL GBM) where the majority of myeloid cells are of monocytic lineage, *Nf1*-silenced murine tumors (resembling human MES GBM) are enriched with neutrophils and brain-resident microglia[4,16], similar to what was shown in human GBM[17]. In fact, multiple subtypes can co-exist within a single tumor, whose relative proportions can evolve over time and/or in response to therapy. Such an example is when tumors with PN signature are treated with radiation (RT) and anti-VEGFA therapy and transition to a MES signature, referred as a PN->MES shift[18]. Other using a combination of high-dimensional technologies, including single-cell RNA sequencing (scRNA-seq) and single-nuclei RNA-seq (snRNA-seq), have documented that GBM contains hierarchies of MES and PN glioma stem cells (GSCs) and their more differentiated progeny both in primary[19] and recurrent tumors[20]. Others explained GBM heterogeneity using the GSC model and showed that they exist along a major transcriptional gradient between two cellular states, developmental and injury response programs[21]. Recurrent GBM showed an increased number of cells with MES signature compared to primary matched samples, suggesting that SOC drives PN->MES transition[20]. This PN-to-MES shift also occurs in our *PDGFB*-driven GBM models in response to the combination of anti-VEGFA and RT[22,23]. Acquisition of the MES signature poses a major clinical challenge as it exemplifies the plasticity of GBM cells and underlines the fundamental problem of treatment resistance to SOC and emerging immunotherapies[20].

Chemokine gradients are essential for hematopoietic-derived myeloid cells to extravasate blood vessels and reach the tumor parenchyma. Monocyte chemoattractant proteins (MCPs) are a group of four structurally related chemokines that are indispensable for monocyte transmigration. In humans, they are encoded by *CCL2 (MCP-1)*, *CCL7 (MCP-3)*, *CCL8 (MCP-2)*, and *CCL13 (MCP-4)* genes that are juxtaposed to each other on chromosome 17; while in mice, MCPs are encoded by *Ccl2*, *Ccl7*, *Ccl8*, and *Ccl12 (MCP-4)* genes clustered on chromosome 11. All MCPs function through engaging the CCR2 receptor, but CCL7 may also interact with CCR1, and CCL13 with CCR3[24]. CCL2 has been found to be critical in promoting the recruitment of monocytes to the CNS[25–28]. Neutralizing monoclonal antibodies against CCL2 had been developed and used in clinical trials against metastatic solid tumors but did not produce favorable outcomes. Meta-analysis of these clinical trials indicated that initial CCL2 inhibition may have unexpectedly caused subsequent increases in circulating CCL2 levels, possibly due to a compensatory feedback loop[29].

In this study, we specifically focused on bone marrow-derived myeloid cells (BMDM) to determine the mechanisms of their infiltration and the role they play in GBM progression. In addition, we wanted to determine whether there is a causal link between various myeloid cell infiltrates and GBM subtype dominance. By creating a combined all MCP-deficient mouse (*qMCP*[−/−]) and generating orthotopic transplant model by injecting PDGFB-driven glioma cells into *qMCP* and wild-type (*WT*) mice we show that loss of expression of all MCPs in the TME results in a decrease of monocyte recruitment and extends survival of tumor-bearing mice. Surprisingly, abolishing all MCPs from the TME and tumor cells together results in compensatory neutrophil recruitment and a shift from PN->MES signature with no effects on the survival of tumor-bearing mice. scRNA-seq and immunohistochemistry reveals that there is an increased presence of Mg and neutrophils in *PDGFB*-driven tumors when MCPs are deleted. Neutrophils are predominantly localized in necrotic areas. Pharmacological targeting of neutrophils and their chemokine receptor CXCR2, or genetic ablation of the neutrophil recruiting chemokine *Cxcl1*, results in extended survival of *PDGFB*-driven tumor-bearing *qMCP*-deficient mice but not *WT* tumor-bearing mice. While abolishing monocyte recruitment in *Nf1*-silenced tumor-bearing mice has no effect on survival of tumor-bearing mice and is not associated with further increase in neutrophil recruitment. In contrast to monocyte abolishment, decreased neutrophil recruitment results in extended survival of *Nf1*-silenced tumor-bearing mice, suggesting that driver mutations not only influence myeloid composition of tumor but also determine their biological function.

Considering GBM contains a mixture of cells with PN and MES gene signatures and their plasticity, these results suggest that effective myeloid therapies should target both neutrophils and monocytes. Since *PDGFB*-driven tumor is mainly monocyte-enriched, we next wanted to determine whether compensatory recruitment of neutrophils is a GBM-specific or CNS-specific phenomenon. To this end, we used a genetic mouse model of hepatocellular carcinoma (HCC). In contrast to *PDGFB*- and *EGFRvIII*-driven murine GBM, and even more so than *Nf1*-silenced GBM, the major immune infiltrates in a genetic HCC mouse model are neutrophils similar what we saw with monocytes in *PDGFB*-driven GBM model. We demonstrate that abolishing monocytes has no impact on the survival of HCC-bearing mice but leads to an increase in the recruitment of neutrophils without affecting survival of tumor-bearing mice. Decreased neutrophil recruitment results in extended survival of HCC tumor-bearing mice. These results suggest that a further increase in already-high infiltration of neutrophils does not accelerate tumor growth, but a decrease in their number improves survival, suggesting a possible plateaued biological effect of neutrophil infiltration in HCC.

In this work, our results suggest there is a compensatory interplay between monocyte and neutrophil recruitment in tumors when tumors are highly enriched with either population. When we target each pro-tumorigenic population separately, we observe compensatory recruitment of the other. Therefore, novel therapeutic strategies should aim at simultaneously targeting both populations to overcome these compensatory recruitment mechanisms.

## Results

### MCPs display region-specific expression patterns, and their increased expression inversely correlates with GBM patient survival

We and others have previously demonstrated that decreased expression of *CCL2* correlates with extended survival of patients with GBM (human GBM; hGBM)[30]. Similarly, mouse GBM (mGBM) models with decreased *Ccl2* expression exhibited prolonged survival relative to *WT*

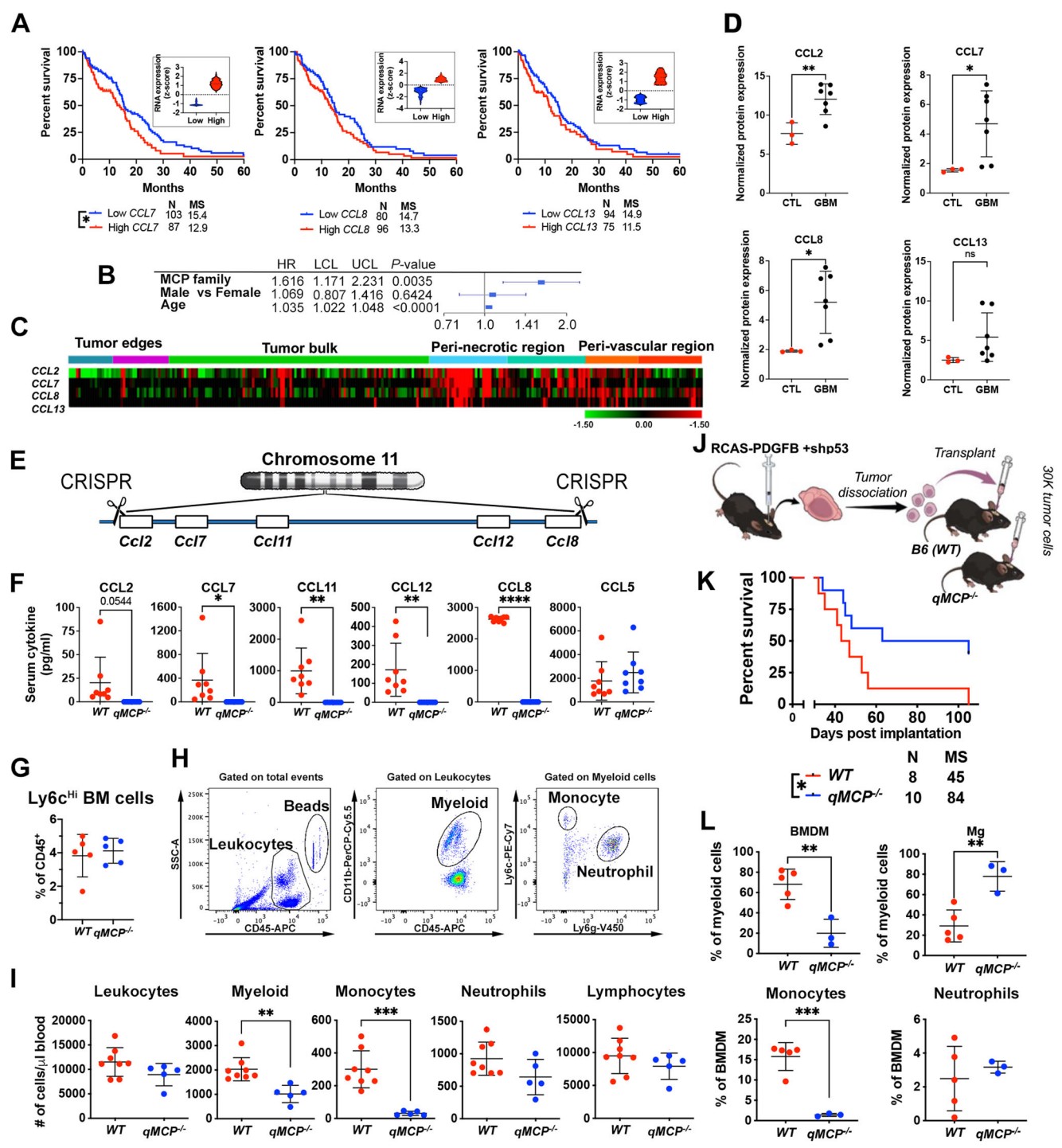

GBM-bearing mice[30,31]. However, in these mice, no decrease in TAM recruitment was observed, suggesting other MCP family members likely compensate for *Ccl2* loss. To determine the basis for this compensatory adaptation, we sought to determine whether the expression of the other MCP members (*CCL7, CCL8, CCL13*) is elevated in hGBM and whether their increased expression serves as a predictor of patient prognosis. We stratified 259 *IDH-WT* patients (with known subtype information) using TCGA datasets using cBioPortal for Cancer Genomics[32,33] into high MCP expressers (+0.5 standard deviation (SD) from the mean of all samples) or low MCP expressers (−0.5 SD from the mean, Fig. 1A, insets) and compared their survival. Patients with elevated *CCL7* expression had a shorter survival time compared to the low expressers (Fig. 1A). A similar trend was also observed with *CCL8* and *CCL13*, although the survival difference was not statistically significant (Fig. 1A). We next selected *IDH-WT* patients with PN and MES expression signature and stratified patients within each subtype into high MCP expressers (+0.5 SD from the mean of all samples) or low MCP expressers (−0.5 SD from the mean, Supplementary Fig. 1) and compared their survival. PN patients with elevated *CCL2, CCL7,* and *CCL13* but not *CCL8* expression had a shorter survival time compared to the low expressers for each of the genes, while in contrast to PN, expression levels of MCPs did not correlate with the survival of the patients with MES signature GBM. Next, we collected the input data from TCGA mRNA expression U133 microarray files. We included a total of 295 *IDH-WT* patient samples for which covariate information (survival information, age, and gender) was available. When controlled for age

**Fig. 1 | Expressions of MCP family member *CCL7* is increased in human GBM, and increased levels correlate with inferior GBM patient survival, and generation and validation of *qMCP*⁻/⁻ mouse. A** Correlations between *CCL7, CCL8,* and *CCL13* expression levels and patient survival were analyzed using an IDH-WT cohort from TCGA. High and low expressions were defined as +/− 0.5 SD from the mean of all samples (n = 259). **B** Forrest plots showing hazard ratio (HR) generated with Cox Proportional Hazards models, using expression of different MCPs genes as continuous variates, with age and gender as covariates. LCL lower confidence limit, UCL upper confidence limit. **C** Expression distribution of *MCP* family members in human GBM tissue as determined in tandem by laser capture microdissection and RNA-seq queried from the IVYGap database. **D** Normalized protein expressions of MCPs examined in control brain and GBM samples by Olink proteomic assay. N = 3 (independent normal controls) and 7 (independent patient tumor tissues). Data are presented as mean +/− SD. **E** Schematic illustration of CRISPR/Cas9-mediated deletion of the *MCP* genes. **F** Serum MCP levels were measured by ELISA following LPS treatment. CCL5 was used as an internal control. N = 8 (independent *WT* mice) and 8 (independent *qMCP*⁻/⁻ mice). Data are presented as mean +/− SD. P = 0.054,

0.0016, 0.0037, 0.000, respectively, where asterisks are present. **G** Flow cytometry quantification of Ly6cᴴⁱ monocytes in the bone marrow of healthy adult mice. Data are presented as mean +/− SD. **H** Multiplex flow cytometry analysis was used to enumerate blood cells in the circulation. N = 5 (independent *WT* mice) and 5 (independent *qMCP*⁻/⁻ mice). **I** Analysis of blood cells in healthy adult mice. P = 0.0018, 0.0003, and 0.0845, respectively, where asterisks are present. N = 8 (independent *WT* mice) and 5 (independent *qMCP*⁻/⁻ mice). Data are presented as mean +/− SD. **J** Schematic illustration of orthotopic transplantation of primary tumors. **K** Kaplan−Meier survival curves of *PDGFB*-driven tumors generated in *WT* and *qMCP*⁻/⁻ mice. P = 0.0333. **L** flow cytometric quantification of myeloid cells in tumors at humane endpoint. N = 5 (independent *WT* mice) and 3 (independent *qMCP*⁻/⁻ mice). P = 0.0040, 0.0048 and 0.0004, respectively, where asterisks are present. Data are presented as mean +/− SD. Two-sided Student's *t* test for (**D, F, G, I, L**), and Log-rank test for (**A, K**). *P < 0.05, **P < 0.01, ***P < 0.001, ****P < 0.0001. MS median survival. CTL adjacent normal brain tissues. Source data are provided as a Source Data file.

and gender, we found that MCP genes collectively serve as a significant adverse prognostic predictor of survival in GBM patients (Fig. 1B). Using the IVYGap (IVY Glioblastoma Atlas Project) database (https://glioblastoma.alleninstitute.org/), we found that *MCPs* are predominantly transcribed in the peri-necrotic and perivascular regions, rather than in the tumor bulk or leading edge (Fig. 1C). This is consistent with the observation that the leading edge is mainly populated by Mg, that does not express CCR2 receptor in tumor[30] and in healthy mice during development and throughout adulthood, therefore does not require MCPs for their infiltration[30,34].

To determine the protein concentration of all MCPs in hGBM samples, we used Olink multiplex proteomics to quantify a predefined group of immune-related proteins (Supplementary Fig. 2). A total of seven *IDH-WT* human GBM tissues along with three normal brain samples were analyzed (sample information in Supplementary Table 1). When MCP expression was specifically assessed, CCL2, CCL7 and CCL8 exhibiting increased levels compared to control brain tissues (Fig. 1D).

**Decreased, but not abolished, *Ccl7, Ccl8,* or *Ccl12* expression leads to extended survival of *PDGFB*-driven GBM-bearing mice**
To investigate the biological significance of the reverse correlation between MCP expressions and survival of GBM patients, we first performed scRNA-seq from *PDGFB*-driven GBM generated in *Ntv-a* mice to determine the sources of MCPs in tumors. We merged all non-neoplastic cells and compared their combined expression of MCP to those of the neoplastic cells. The results show that both TME cells and tumor cells contribute to comparable amounts of the total MCP pool (Supplementary Fig. 3A). Next, we leveraged existing genetically modified mice that are knocked out of individual MCPs. In these experiments, we orthotopically transplanted *PDGFB*-driven primary mGBM tumor cells into the brains of wild-type (*WT*) mice and mice deficient in *Ccl7* or *Ccl8/12* expression (Supplementary Fig. 3B), a scheme similar to we have shown previously for *Ccl2*[30]. It should be noted that *Ccl7* or *Ccl8/12* are depleted in the TME but are retained in tumor cells. While we observed increased survival in these tumor-bearing mice relative to *WT* controls, there was no reduction in tumor-associated macrophage content by IBA1⁺ (a pan-macrophage marker, which is expressed by monocytes, MDM and Mg, Supplementary Fig. 3C) areas. Based on this finding, we wondered whether complete genetic deletion of MCPs from both tumor cells and the TME could further extend the survival of tumor-bearing mice. To address this question, we induced de novo *PDGFB*-driven tumors in *WT;Ntv-a; Ccl2*⁻/⁻*; Ntv-a, Ccl7*⁻/⁻*; Ntv-a,* and *Ccl8/12*⁻/⁻*; Ntv-a* mice by injecting a combination of RCAS-*shp53* and RCAS-*PDGFB* in the frontal striatum. Unexpectedly, the survival benefits previously observed with

the transplant model were abolished when these MCPs were individually deleted in both the tumor cells and the TME (Supplementary Fig. 3D). When we examined the immune composition of the tumors by flow cytometry, using *Ccl8/12*⁻/⁻*; Ntv-a* mice to represent the entire cohort, no difference was observed in the proportion of infiltrating monocytes or Mg compared to *WT; Ntv-a* controls, although the decrease in BMDM was associated with an increase in Mg (Supplementary Fig. 3E). These results suggest that partial loss of MCPs, which does not trigger a compensatory response, provides a survival advantage, but complete deletion of an individual MCP may cause other MCP members to compensate.

**Creating *quintuple MCP-KO* mice using CRISPR/Cas9**
Because of the functional redundancy of the MCP members, we sought to generate a knockout (*KO*) strain devoid of all MCPs by interbreeding each individual *MCP-KO* line (*Ccl2*⁻/⁻ X *Ccl7*⁻/⁻ X *Ccl8/12*⁻/⁻). However, this approach proved futile, due to the close linkage of *Ccl2, Ccl7, Ccl8,* and *Ccl12* on chromosome 11, making homology recombination unfeasible. To surmount this obstacle, we designed a strategy to collectively delete all the MCP genes using CRISPR/Cas9-based technology. Combined, these genes span only ~80k base pairs on chromosome 11 (Fig. 1E). *Ccl11* (Eotaxin) was also deleted because it intercedes the MCPs (Fig. 1E). These *quintuple knockout* (*qMCP*⁻/⁻) adult mice were then validated by lipopolysaccharide (LPS, IP, 1 mg/kg) injection. Each individual MCP was quantified in the serum by ELISA, demonstrating an absence of all (Fig. 1F). We used CCL5 as a positive control, whose gene is also located on chromosome 11, but is further away from the MCPs (Fig. 1F). Next, we performed extensive characterization of the brain (Supplementary Fig. 4), bone marrow (Supplementary Fig. 5), and spleen (Supplementary Fig. 6) of healthy non-tumor-bearing adult *qMCP*⁻/⁻ mice by flow cytometry. We did not observe any differences in Mg (Supplementary Fig. 4B) or bone marrow monocytes (Fig. 1G and Supplementary Fig. 4C), but noted an increase in neutrophils in bone marrow (Supplementary Fig. 5B) and a reduction in total monocytes and Ly6cᴴⁱ monocytes in the spleen (Supplementary Fig. 6B). It is well known that in physiological conditions, extravasated Ly6cᴴⁱ monocytes can be found in almost all tissues throughout the body, where they constitute a minor yet significant fraction of the tissue macrophage pool[35]. Exceptions include brain[36,37], the alveolar macrophage pool in the lung[38], and epidermis[39,40]. The spleen serves as a significant peripheral reservoir for Ly6cᴴⁱ and Ly6cᴸᵒ monocytes. These cells can maintain their monocyte-like state and mirror their blood counterparts' expression profiles and morphology[41]. The origin of splenic Ly6cᴴⁱ and Ly6cᴸᵒ monocytes has been actively debated in the field, and it remains a question of whether, like what was shown

for blood, Ly6c$^{Hi}$ monocytes can give rise to Ly6c$^{Lo}$ monocytes[42]. In this regard, our data show that when monocytes from BM are blocked, the number of Ly6C$^{Hi}$ as well Ly6c$^{Lo}$ monocytes are significantly impaired in the spleen, suggesting that BM is a significant source of Ly6c$^{Hi}$ monocytes and raising the possibility that, similar to what was shown in blood, Ly6c$^{Hi}$ monocytes are progenitors for Ly6c$^{Lo}$ monocytes in the spleen. When we analyzed the absolute count of leukocytes in the blood by flow cytometry (Fig. 1H), there were reduced CD11b$^+$ myeloid cells, likely attributable to the loss of Ly6c$^{Hi}$ inflammatory monocytes (Fig. 1I). No difference in blood neutrophils or lymphocytes was observed (Fig. 1I).

## Genetic deletion of qMCP results in a compensatory influx of neutrophils

Leveraging this mouse strain, we next sought to determine the role of stroma-derived MCPs in tumor monocyte recruitment. For these studies, we generated GBMs in WT; Ntv-a mice with RCAS-shp53 and RCAS-PDGFB. When tumors emerged, freshly dissociated tumor cells were orthotopically transplanted into the brains of qMCP$^{-/-}$ and WT (B6) mice (Fig. 1J). Kaplan−Meier analysis demonstrated that eliminating all MCPs from the stroma extended the survival time of tumor-bearing qMCP$^{-/-}$ mice (Fig. 1K). FACS analysis showed decreased BMDM infiltration, which likely resulted from decreased Ly6c$^{Hi}$ monocytes (Fig. 1L). When compared to the results using single chemokine KO mice (Supplementary Fig. 3B), where no reduction of infiltrating monocytes was observed, these results suggest that all MCP members contribute to monocyte recruitment and that loss of one member can be compensated by other MCPs. Immunohistochemistry for IBA1 was used to complement the flow cytometry results (Supplementary Fig. 7). We did not observe a difference between these two genotypes with IBA1 quantification. Because IBA1 visualizes both monocyte, MDM as well as Mg, when monocyte infiltration is reduced in qMCP$^{-/-}$ mice, an anti-parallel increase in Mg occurs; therefore, the overall number for combined monocyte/MDM/Mg positive area remains unchanged between tumors generated in WT and qMCP$^{-/-}$ mice.

To determine whether MCPs have direct impact on tumor cells, we generated PDGFB-driven GBM in WT;Ntv-a mice and derived primary tumor cell cultures and maintained them in stem cell enriching medium− where we refer to them as WT primary glioma stem cell cultures (WT-GSCs) (Supplementary Fig. 8A), since we have previously shown that in these condition tumor cells retain better original tumor cell properties and exhibit stem cell properties[43–45]. Further, we stimulated WT-GSCs with increasing doses of recombinant MCPs and evaluated their growth by MTS assay. We did not observe changes in tumor cell growth (Supplementary Fig. 8B), suggesting that MCPs do not affect tumor growth.

To determine whether survival is extended when all MCPs are genetically ablated in both TME and tumor cells, we induced de novo GBM in WT; Ntv-a and qMCP$^{-/-}$; Ntv-a mice by co-injecting RCAS-shp53 and RCAS-PDGFB in the frontal striatum. We hypothesized that abolishing MCPs would inhibit monocyte tumor infiltration, thereby extend the survival of GBM-bearing mice. Surprisingly, there was no difference in survival between WT; Ntv-a and qMCP$^{-/-}$; Ntv-a tumor-bearing mice (Fig. 2A). To understand the cellular and molecular mechanisms underlying this unexpected result, we analyzed WT and qMCP-deficient tumors by single-cell RNA sequencing (scRNA-seq, Supplementary Fig. 9). After filtering out low-quality cells and putative doublets ("Methods", Supplementary Fig. 9A−D), we performed unsupervised clustering on 57,360 cells and identified five major cell classes−lymphoid, myeloid, stromal, endothelial, and malignant (Fig. 2B). Within each class we further stratified cells into phenotypical or functional subsets according to their unique gene signatures (Fig. 2B and Supplementary Fig. 9E, F). MCP transcripts can be detected in many cell types in WT; Ntv-a mice, particularly malignant cells, MDM, and

monocytes, but were undetectable in qMCP$^{-/-}$; Ntv-a mice, reaffirming the efficacy of gene deletion (Supplementary Fig. 10). In addition, we found a decrease in monocytes in qMCP$^{-/-}$; Ntv-a mice, consistent with this genotype and a corresponding increase of neutrophil infiltration in qMCP$^{-/-}$; Ntv-a mice (Fig. 2C). By UMAP dimensionality reduction of the neutrophil population, one distinct cluster was identified in WT and MCP-deficient tumors (cluster 2). In contrast, two additional neutrophil clusters were present (cluster 0 and cluster 1) primarily in MCP-deficient tumors (Fig. 2D, left panel). Neutrophils in cluster 2 show an elevated abundance of Arg1, Cd74, H2-Aa, Ctsc (gene coding for cathepsin C) transcripts among others (Supplementary Fig 11A). Cluster 1 shows an elevated abundance of Cd274 (gene coding PD-L1 protein), Ccl3, Cxcl3, and Cstb (gene coding for cathepsin B) (Supplementary Fig. 11A). This cluster resembles the immunosuppressive neutrophil subgroup also expressing the neutrophil recruitment chemokine Cxcl3. Cluster 0 shows an elevated expression of Cxcr2, Trem1, Il1b, Csf3r2 and resembles pro-inflammatory neutrophils (Supplementary Fig. 11A). These results suggest that both neutrophil abundance and functional gene expression programs are altered in qMCP-deficient tumors, where two novel subsets emerged.

We next evaluated MDM population and identified four clusters in WT and MCP-deficient tumors (Fig. 2D, right panel). Cluster 1 shows an enhanced expression of Apoe, Trem2, Ctsd (Supplementary Fig. 11B). Cluster 2 shows an elevated expression of cell cycle markers Hist1h2ae, Tubb5, Mki67 (coding for Ki67 protein), and Hmgb2 (Supplementary Fig. 11B) and is enriched for proliferating MDMs. Cluster 3 expresses Mrc1 and Cd163, typically found in alternatively activated macrophages (Supplementary Fig. 11B). Cluster 0 was substantially decreased in qMCP-deficient tumors. Cells in this cluster express Il1b, Ccr2, Fos, and Junb (Supplementary Fig. 11B). We did not observe any substantial differences in Mg subcluster cell composition between the two genotypes (Supplementary Fig. 11C).

To complement and corroborate the scRNA-seq data, we used multi-parameter spectral flow cytometry to analyze the composition of myeloid cells in these tumors (Fig. 2E). Based on the combination of multiple surface markers (gating strategy shown in Supplementary Fig. 12), we identified total myeloid cells (CD11b$^+$CD45$^+$), which comprise of both brain-resident Mg (CD11b$^+$CD45$^{Lo}$Ly6c$^{Neg}$Ly6g$^{Neg}$CD49d$^{Neg}$) and infiltrating bone marrow-derived myeloid cells (CD11b$^+$CD45$^+$CD49d$^+$). These infiltrating myeloid cells were further stratified into inflammatory monocytes (CD11b$^+$CD45$^{Hi}$Ly6c$^{Hi}$Ly6g$^{Neg}$CD49d$^+$), monocyte-derived macrophages (CD11b$^+$CD45$^{Hi}$Ly6c$^{Lo/Neg}$Ly6g$^{Neg}$CD49d$^+$F4/80$^+$), or neutrophils (CD11b$^+$CD45$^+$Ly6c$^+$Ly6g$^+$CD49d$^+$, Supplementary Fig. 12). Quantitatively, we did not observe any difference in total myeloid cells, Mg, or BMDM between the two genotypes (Supplementary Fig. 12B). These results were further confirmed by quantification of immunohistochemistry using Mg-specific anti-P2YR$_{12}$ antibody (Supplementary Fig. 13). However, we found a reduction in the presence of Ly6c$^{Hi}$ inflammatory monocytes and an increase in Ly6g$^+$ neutrophils in tumors in qMCP$^{-/-}$;Ntv-a mice (Fig. 2F). Moreover, in-depth analyses of the lymphocyte compartment (Foxp3$^+$ Treg cells, exhausted CD8$^+$ T cells, B cells, and NK cells, Supplementary Fig. 14) and dendritic cells (DCs, DC1 and DC2, Supplementary Fig. 15) did not reveal any differences between WT;Ntv-a and qMCP$^{-/-}$;Ntv-a mice. To further confirm that the decreased recruitment of Ly6c$^{Hi}$ monocytes leads to increased neutrophils as a result of the effect of qMCPs on the inflammatory monocyte CCR2 receptor, and that the loss of qMCPs does not affect Mg, we orthotopically transplanted freshly dissociated PDGFB-driven GBM cells into Cx3Cr1$^{GFP/WT}$;Ccr2$^{RFP/WT}$ double heterozygous reporter knock-in and Cx3Cr1$^{GFP/WT}$;Ccr2$^{RFP/RFP}$ (heterozygous knock-in for Cxc3Cr1 gene and homozygous knock-in for Ccr2) mice. Using PDGFB-driven GBM in Cx3Cr1$^{GFP/WT}$;Ccr2$^{RFP/WT}$ mice, we previously demonstrated that CX3CR1$^{Lo}$CCR2$^{Hi}$ monocytes infiltrate GBM and transition to CX3CR1$^{Hi}$CCR2$^{Lo/neg}$ MDM and together constitute nearly 85% of the tumor-associated macrophages; while CX3CR1$^{Hi}$CCR2$^{neg}$ Mg only

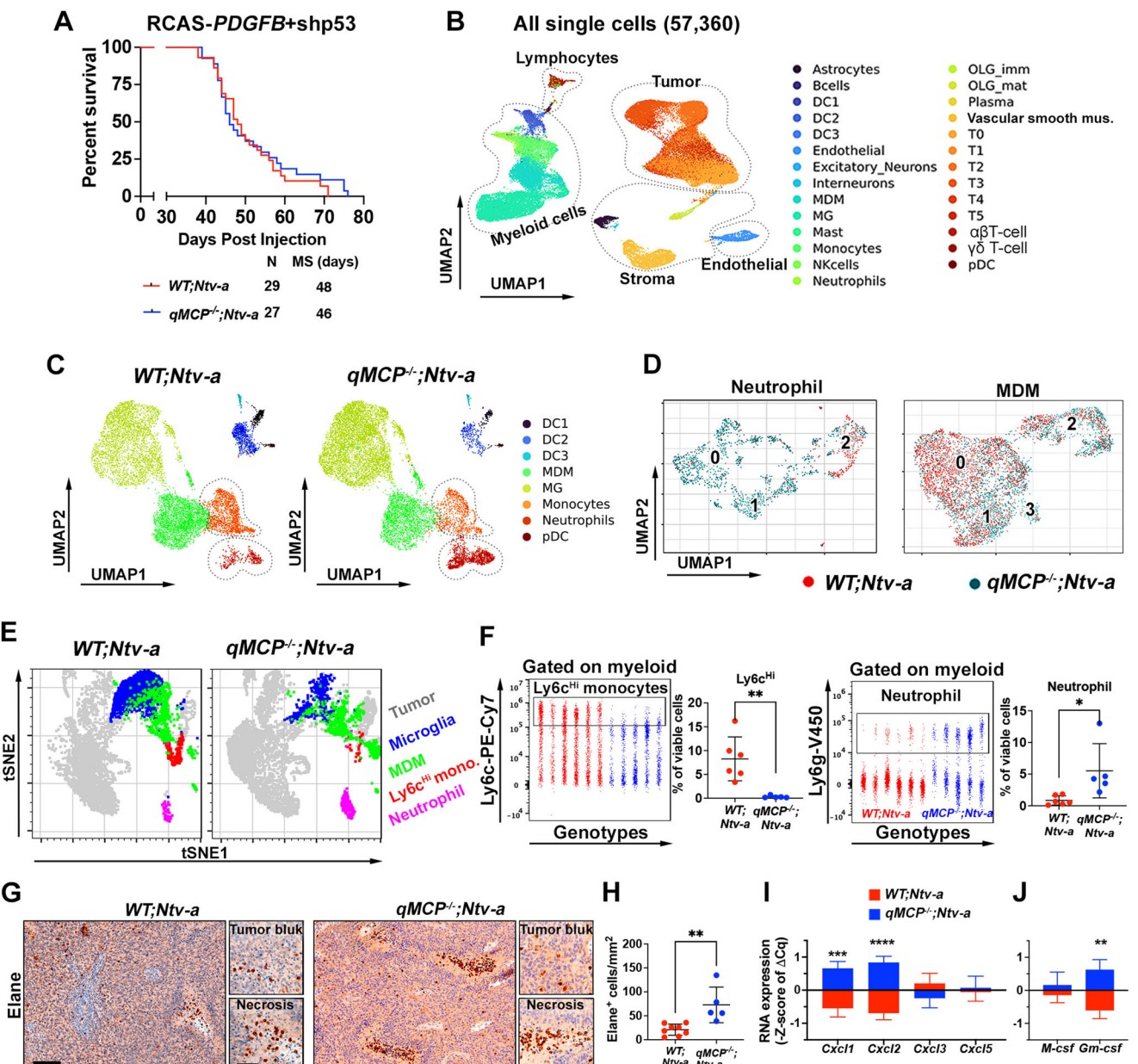

**Fig. 2 | Deletion of MCPs blocks monocyte recruitment and leads to a compensatory infiltration of neutrophils. A** Kaplan–Meier survival curves of *PDGFB*-driven tumors generated in *WT; Ntv-a* and *qMCP⁻/⁻; Ntv-a* mice. **B** UMAP dimensionality reduction of scRNA-seq data from 57,360 cells isolated from three *WT; Ntv-a* and three *qMCP⁻/⁻; Ntv-a* tumors. Consistent expression of known markers was used to annotate cell clusters into five broad cell classes: lymphoid (B cells, NK cells, plasma, αβT-cell, and γδT cells), Myeloid (DC1, DC2, DC3, pDC, MDM, MG, monocytes and neutrophils), Stromal (astrocytes, excitatory neurons, interneurons, OLG_imm, OLG_mat and vascular smooth muscle cells), endothelial, and tumor (T0 to T5). **C** UMAP showing refined clustering of myeloid cells isolated from *WT;Ntv-a* (left) and *qMCP⁻/⁻ ; Ntv-a* (right) tumors. **D** UMAP dimensionality reduction of neutrophil and MDM from qMCP-deficient and WT tumors into clusters. **E** tSNE plots showing results of spectral flow cytometry analysis of the tumors. **F** Dot plots

and quantification of monocytes and neutrophils analyzed by spectral flow cytometry. *P* = 0.0040 and 0.0263, respectively. *N* = 6 (independent *WT; Ntv-a* mice) and 5 (independent *qMCP⁻/⁻ ; Ntv-a* mice). Data are presented as mean +/− SD. **G** Representative images of Immunohistochemistry staining for Elane in GBM sections, and **H** quantification of Elane⁺ cells. *P* = 0.0033. *N* = 8 (independent *WT;Ntv-a* mice) and 5 (independent *qMCP⁻/⁻;Ntv-a* mice). Data are presented as mean +/− SD. **I, J** Quantitative analysis of *Cxcl* chemokines or *Csf* by qPCR. Centerline shows Z score equals to 0, whiskers indicate the standard deviation. *N* = 12 (independent *WT;Ntv-a* mice) and 10 (independent *qMCP⁻/⁻;Ntv-a* mice). Two-tailed Student's *t* test for (**F, H, I, J**), and Log-rank test for (**A**). **P* < 0.05, ** *P* < 0.01, ****P* < 0.001, *****P* < 0.0001. Scale bar = 100 µm, scale bar in inset = 50 µm. Mus muscle, imm immature, mat mature. Source data are provided as a Source Data file.

account for 15% of the myeloid cells and occupy mostly peritumoral areas[30,46]. This finding was later confirmed by tracing each subset using single-cell resolution 2-photon live-imaging[46]. Multi-parameter spectral flow cytometry used to analyze the composition of myeloid cells in tumors generated in *Cx3Cr1^GFP/WT; Ccr2^RFP/RFP* and *Cx3Cr1^GFP/WT;Ccr2^RFP/RFP* mice showed that, similar to *PDGFB*-driven tumors generated in *qMCP⁻/⁻;Ntv-a* mice, there was a reduction in monocyte and MDM and an

increase in Ly6g⁺ neutrophils (Supplementary Fig. 16A, B). Similar results were shown recently when RT was performed in *CCR2*-deficient medulloblastoma-bearing mice where compensatory neutrophil recruitment occurred, resulting in no beneficial effect on tumor progression, although the biological significance of neutrophil influx was not evaluated[47]. Quantification of histological sections of tumors from *Cx3Cr1^GFP/WT;Ccr2^RFP/RFP* and *Cx3Cr1^GFP/WT;Ccr2^RFP/RFP* mice showed that *Ccr2*

loss reduces monocyte/MDM infiltration but has no impact on Mg in the peritumoral areas (Supplementary Fig. 16C, D). To confirm that CX3CR1+CCR2- positive cells in the perivascular areas were Mg, we stained the sections for TMEM119, a Mg-specific marker (Supplementary Fig. 16E). These results using Ccr2-deficient mice demonstrate that qMCPs signal in GBM through the CCR2 receptor expressed in monocytes, and that Ly6c^hiCCR2+ inflammatory monocyte abolishment in PDGFB-driven tumors is associated with an influx of Ly6g+ neutrophils.

Next, to confirm this neutrophil infiltration and spatially resolve their presence in tumor tissue, we used immunohistochemistry staining of a neutrophil-specific elastase (Elane; Fig. 2G). We found increased neutrophils in qMCP−/−;Ntv-a mice (Fig. 2H), consistent with the scRNA-seq and spectral flow cytometry results. Interestingly, these neutrophils tended to cluster around or within the necrotic regions (Fig. 2G), similar to what was recently reported by others[48]. To determine whether the increased neutrophil influx was associated with increased levels of neutrophil recruitment chemokines, we performed qPCR for Cxcl1, Cxcl2, Cxcl3, and Cxcl5, which are known to mobilize and recruit neutrophils to site of inflammation in the CNS[49]. RNA-expression analysis demonstrates significant increases in Cxcl1 and Cxcl2 expressions in tumors from qMCP−/−; Ntv-a mice compared to WT; Ntv-a mice (Fig. 2I). Since Cxcl1 is a major neutrophil recruitment chemokine in mice, these data suggest that increased Cxcl1 levels may be responsible for the increased neutrophil content in qMCP-deficient tumors. In addition, we detected increased granulocyte-macrophage colony-stimulating factor (Gm-csf), a cytokine essential for neutrophil development and survival[50], expression in qMCP−/−; Ntv-a mice, but not macrophage colony-stimulating factor (M-csf) (Fig. 2J). To define the cellular source of neutrophil recruitment chemokines and Gm-csf, we analyzed the scRNA-seq data (Supplementary Fig. S17). We found that these chemokines are predominantly expressed by endothelial cells or by the neutrophils themselves (Supplementary Fig. 17A). Interestingly, loss of MCPs increased Cxcl2 and Cxcl3 expression by neutrophils. It has been shown that increased expression of Cxcl2 by neutrophils can increase their recruitment in both autocrine and paracrine fashions during cutaneous infection[51], which raise the possibility that, in the context of qMCP loss, neutrophils facilitate their own recruitment into GBM, which can explain the influx of neutrophils in qMCP−/− mice compared to WT controls. Further, similar to the Ccl chemokines, these Cxcl chemokines are produced by both malignant and non-malignant cells, although non-malignant cells are larger contributors (Supplementary Fig. 17B). Taken together, abrogating MCP expression results in a near-complete blockade of tumor monocyte recruitment and a compensatory influx of neutrophils, which is associated with increased expression of Cxcl, Cxcl2 and Gm-csf.

While CSF1R inhibition in PDGFB-driven GBM-bearing mice[52] is ineffective as a monotherapy, the CSF1R inhibitor BLZ945 showed a synergistic effect when combined with radiation (RT)[53]. To determine whether abolishing monocytes in combination with RT can also increase anti-tumor efficacy, WT; Ntv-a and qMCP−/−; Ntv-a tumor-bearing mice were treated with RT (Supplementary Fig. 18). Interestingly, RT did not provide survival advantage to PDGFB-driven tumors in qMCP−/−; Ntv-a mice compared to WT;Ntv-a. These different outcomes between CSF1R inhibition and MCP abolition can be partially attributed to the inefficacy of CSF1R inhibitor in decreasing tumor-associated macrophage numbers in GBM; therefore, no compensatory recruitment of neutrophils would be present. The results described here also suggest that compensatory neutrophil recruitment in monocyte-abolished tumors may reverse the RT efficacy, similar to what was recently shown that locally activated neutrophils as a result of irradiation can create a tumor-supportive microenvironment in the lungs[54].

## Human MES tumors show the increased neutrophil presence

Increased neutrophil influx is a prominent feature in qMCP−/−;Ntv-a mice, reminiscent of the mesenchymal (MES) hGBM subtype, which have an abundance of neutrophils[16]. Using NanoString Pan-Cancer Immune Pathways analyses of hGBM with known molecular subtypes (determined by custom-made probes for 152 genes from the original GBM_2 design)[15,55,56], we dissected the cellular landscape of IDH-Mut (G-CIMP) and IDH-WT samples (including PN, CL and MES, Fig. 3A). In silico deconvolution of these data showed an increased "neutrophil score" in MES human GBM relative to all other molecular subtypes (Fig. 3B). Consistent with this finding, the neutrophil recruiting chemokine CXCL8 and their characteristic surface marker S100A9 were elevated in MES hGBM, although the latter was not statistically significant (Fig. 3C).

To extend our discovery to a larger cohort of GBM patients, we culled data from TCGA (cBioportal, Firehose Legacy, 273 GBM patient samples with mutation and copy number alteration (CNA) data) and filtered the samples using criteria so that they only present alterations in one of the following two driver genes— PDGFRA (driver of the PN subtype) or NF1 Del/Mut (MES subtype, Fig. 3D) and lack EGFR alteration. Then we performed comparisons between PDGFRA- and NF1-altered patient tumors (Fig. 3D). Similar to the observations above, NF1-altered tumors, which predominantly cluster in MES subtype, showed significantly higher expression of neutrophil recruitment chemokines (Cxcl-1, −2, −5, −8) compared to PDGFRA-altered tumors, which predominantly cluster in the PN expression signature group[7,10]. No difference was observed in chemokine receptor Cxcr2 (Fig. 3E). Interestingly, when we stratified GBM patients based on their expression of CXCL8 (IL8, 373 patients with detectable IL8), a potent neutrophil recruitment chemokine in humans (same method as described in Fig. 1A), increased expression of CXCL8 was associated with reduced patient survival (Fig. 3F), analogous to prior reports[16].

Finally, to determine whether increased neutrophil recruitment chemokines exist at the protein level, we examined their concentrations in human GBM samples by Olink® proteomic assay (Supplementary Fig. 19). Using a total of 3 IDH-WT MES human GBM tissues and three normal brain samples, we found an increase in chemokines involved in neutrophil recruitment. Furthermore, when we analyzed subtype-defined human GBM samples[15] by immunohistochemistry for Elane expression (Fig. 3G), there was a similar increase in the number of neutrophils in MES tumors relative to IDH-Mut, PN, and CL GBM samples (Fig. 3H).

## Increased neutrophil influx leads to a tumor transition from PN to MES signature

When GBMs with a PN signature are treated with SOC, they often recur with a MES signature, referred to as PN to MES transition (PN- > MES transition)[18,20]. This shift also occurs in PDGFB-driven mGBM models in response to anti-VEGFA or RT[22,23]. It is interesting to note that neutrophils are highly enriched in qMCP−/−;Ntv-a mice-bearing tumors generated with PDGFB overexpression, a potent driver mutation of PN GBM. Together, these observations prompted us to determine whether increased neutrophil infiltration can induce PN- > MES transition, especially in light of a recent study revealing that reciprocal interactions between tumor-associated macrophages (includes MDM and Mg) and tumor cells can drive the transition of GBM to a MES-like state[14,57]. We used a previously published method for classifying human GBM cells into distinct and reproducible cellular states (MES-like, AC-like, NPC-like, and OPC-like)[13,14], and demonstrate that they are present in our scRNA-seq datasets of PDGFB-driven tumors generated in WT; Ntv-a and qMCP−/−; Ntv-a mice, although with a reduction of OPC- and NPC-like cells and increase in MES-like cells in MCP-deficient tumors compared to WT (Fig. 4A). Next, we analyzed the malignant cells identified in our scRNA-seq datasets for their MES-like state expression score (Fig. 4B). We found a significant increase in MES-like scores in qMCP−/−; Ntv-a mice relative to WT;Ntv-a mice (Fig. 4C), suggesting the tumor cells were undergoing PN- > MES transition. To substantiate this result, we leveraged a previously published qPCR panel[58] that includes genes associated with either PN or MES subtypes, some of which (e.g., SERPINE1, CHI3L1)

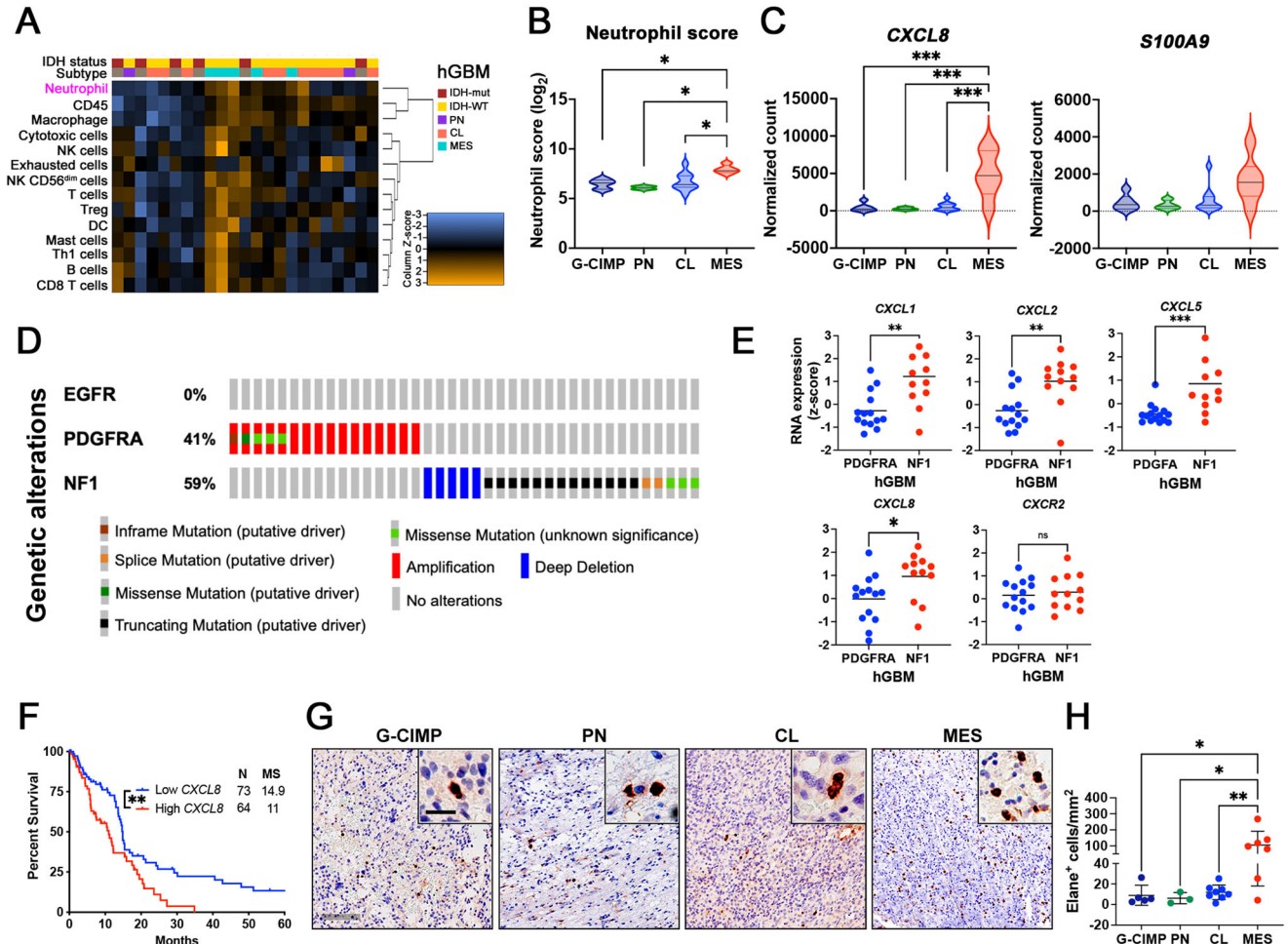

**Fig. 3 | Human MES and NF1 Del/Mut GBM have increased expression of neutrophil recruitment chemokines and neutrophil content. A** NanoString in silico analysis of cellular scores in human GBM tumor samples. ($N = 6$ for G-CIMP or IDH-mut, 2 for PN, 10 for CL, and 5 for MES GBM patients). **B** Neutrophil score in hGBM subtype samples. $P = 0.0105, 0.0212$, and $0.0191$, respectively. $N$ is the same as in (**A**). **C** Expression of neutrophil recruitment chemokines *IL8* and *S100A9* examined by NanoString. $P = 0.0003, 0.0007$, and $0.0001$, respectively. $N$ is the same as in (**A**). **D** Genetic alterations of GBM patient samples (cBioportal, TCGA, Firehose Legacy) selected based on mutual exclusivity of alterations in *PDGFFRA* ($N = 15$), *NF1* ($N = 23$), and *EGFR* ($N = 0$). **E** Expression of neutrophil recruitment chemokines and their shared receptor CXCR2 examined by TCGA. Two-sided Student's *t* test. $N = 14$ (independent *PDGFRA*-amplified patients) and 11 (independent *Nf-1* silenced patients). Data are presented as mean +/− SD. **F** Survival curves of *IDH-WT* human GBM patients based on low- and high- expression levels of *IL8*. High and low are defined as +/− 1 SD from mean of 373 *IDH-WT* GBM patient samples (cBioportal, TCGA, Firehose Legacy). $P = 0.0029$ by Log-rank test. **G** Representative images of IHC for Elane. **H** Quantification of Elane+ neutrophils. $N = 5$ (independent G-CIMP patients), 3 (independent PN patients), 8 (independent CL patients), and 7 (independent MES patients). Data are presented as mean +/− SD. $P = 0.0170, 0.0411$, and $0.0084$, respectively where asterisks are present. One-way ANOVA with Tukey's multiple comparisons test for (**B, C, H**), two-sided Student's *t* test for (**E**), Log-rank test for (**F**). *$P < 0.05$, **$P < 0.01$, ***$P < 0.001$. Scale bar = 100 μm, scale bar in inset = 25 μm. MS median survival. Source data are provided as a Source Data file.

overlap with the Hara et al. dataset[14]. As predicted by the observed PN->MES transition, we found increases in many of these MES-related genes (*MGMT, SERPINE1, TAZ, CASP1, TGFB*) in *qMCP−/−; Ntv-a* mice (Fig. 4D).

Next, we analyzed OLIG2, GFAP, and CD44 expression by immunohistochemistry (Fig. 4E), and found no difference in OLIG2 expression between these two genotypes; however, both GFAP and CD44, canonical markers of MES hGBM were increased in *qMCP−/−;Ntv-a* mice (Fig. 4E). Taken together, we establish that neutrophil tumor infiltration following monocyte abolition induced PN->MES transition.

In addition to the molecular changes observed within the tumor tissue, we sought to determine whether *MCP* loss changes the cellular composition in the tumor microenvironment (Fig. 4F). IBA1, a pan-macrophage marker, labels tumor-associated macrophages (TAM) regardless of their origin (Mg, monocytes, and MDM). The IBA1-positive areas within the core of the tumors were decreased in tumors generated in *qMCP−/−; Ntv-a* mice compared to *WT; Ntv-a* mice, as expected for this genotype (Fig. 2E). We also used the Mg-specific

marker P2YR12 to demonstrate increased Mg content at the tumor margins in *qMCP−/−; Ntv-a* mice (Fig. 4F). No differences in blood vessel sizes (CD31 reactivity; Fig. 4F) were observed.

### Defining the molecular mechanisms underlying the pro-tumor neutrophil effects in GBM

To study the molecular mechanism(s) underlying the tumor-promoting effects of neutrophils in *qMCP−/−;Ntv-a* mice, we performed weighted gene co-expression network analysis (WGCNA) on all the malignant cells detected in our scRNA-seq data (Supplementary Fig. 20). This analysis revealed gene regulatory network, or "modules", based on gene co-expression patterns[59], enabling us to identify co-regulated genes shared across multiple cell clusters. Among all the modules examined, the "Greenyellow" module showed a prominent increase in *qMCP−/−; Ntv-a* mice as quantified by its module score (Fig. 5A). The most co-expressed genes in this module form an interconnected graph (Fig. 5B) and consist of genes implicated in glycolysis

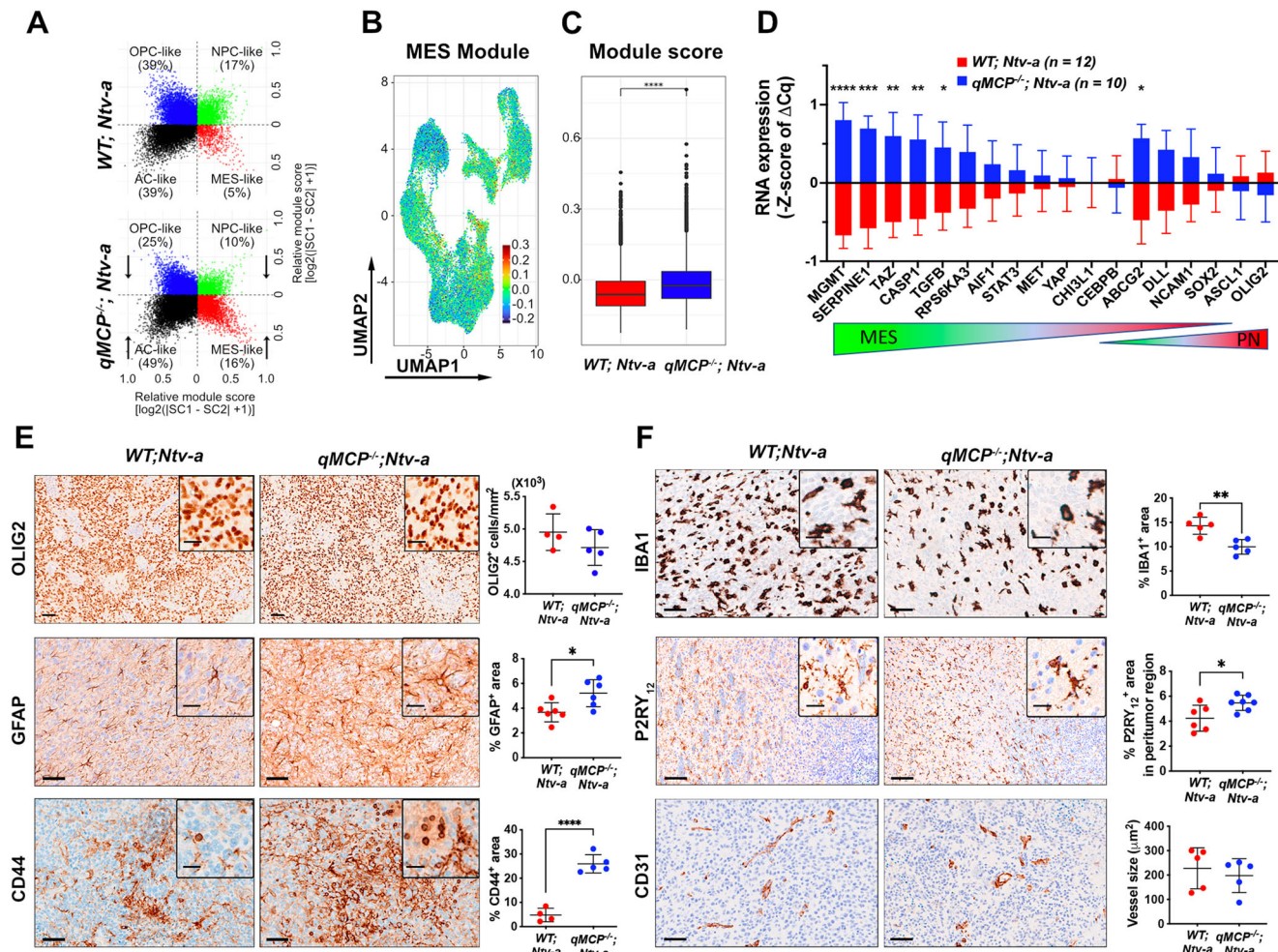

**Fig. 4 | Deletion of MCP chemokines results in PN to MES shift in *PDGFB*-driven GBM. A** Neftel cell state plots from scRNA-seq demonstrate *PDGFB*-driven tumors in absence of monocyte show decrease in OPC- and NPC-like cells and increase in MES-like cells. **B** UMAP dimensionality reduction of MES-like cell state module score in all malignant cells examined by scRNA-seq. **C** Quantification of MES score between *WT; Ntv-a* (red) and *qMCP⁻/⁻; Ntv-a* (blue) malignant cells. Two-tailed Wilcoxon signed-rank test was used on the cell distributions. No adjustment was made. **D** Real-time qPCR panel of signature genes that are differentially expressed in PN and MES human GBM. Centerline shows Z score equals to 0, whiskers indicate standard deviation. Data are presented as mean +/− SD. P = 0.0000, 0.0007, 0.0052, 0.0110, 0.0432, and 0.0108, respectively where asterisks are present. N = 12 (*WT; Ntv-a*) and 10 (*qMCP⁻/⁻; Ntv-a*) mice. **E** Representative images and quantification of

immunohistochemistry for signature molecules of neoplastic cells. Data are presented as mean +/− SD. P = 0.0183 and 0.0000, respectively, where asterisks are present. N = 4 (*WT; Ntv-a*) and 5 (*qMCP⁻/⁻; Ntv-a*) mice for Olig2; 6 (*WT; Ntv-a*) and 6 (*qMCP⁻/⁻; Ntv-a*) mice for GFAP, 4 (*WT;Ntv-a*) and 5 (*qMCP⁻/⁻; Ntv-a*) mice for CD44. **F** Representative images and quantification of immunohistochemistry for molecules in TME. N = 5 (*WT; Ntv-a*) and 5 (*qMCP⁻/⁻; Ntv-a*) mice for IBA1; 6 (*WT; Ntv-a*) and 7 (*qMCP⁻/⁻; Ntv-a*) mice for P2RY12, 5 (*WT; Ntv-a*) and 5 (*qMCP⁻/⁻; Ntv-a*) mice for vessel size comparison. Data are presented as mean +/− SD. P = 0.0030 and 0.0225, respectively, where asterisks are present. Two-sided Wilcoxon signed-rank test for (**C**). Two-sided Student's *t* test for (**D–F**). *P < 0.05, **P < 0.01, ***P < 0.001, ****P < 0.0001. Scale bar = 50 μm, scale bar in inset = 20 μm. Source data are provided as a Source Data file.

(*Gapdh, Pgk1, Pgam1*) and hypoxia (*Aldoa, Mif, Ldha*). To determine the biological functions of WGCNA modules, we performed pathway enrichment analysis using the Hallmark gene sets (Fig. 5C). Among others, we found significant enrichment in glycolysis, hypoxia, and mTOR signaling in the "Greenyellow" module (arrowheads, Fig. 5C), indicating that the tumor cells in *qMCP⁻/⁻; Ntv-a* mice are likely experiencing higher metabolic stress. These findings were also recapitulated using GO (gene ontology) pathways (Supplementary Fig. 21A). Interestingly, when we assessed glycolysis in WT-GSCs and qMCP⁻/⁻-GSCs in vitro, we did not observe a difference between these two genotypes, regardless whether they were grown in normoxic (Supplementary Fig. 21C) or hypoxic (Supplementary Fig. 21D) conditions, indicating that cellular interactions between neoplastic cells and its microenvironment are likely essential for inducing this metabolic stress.

To determine how neutrophils contribute to the metabolic changes in tumors, we performed ligand-receptor interaction inference using CellPhoneDB (v2.0)[60]. CellPhoneDB predicts interactions between two cell types based on the coordinated expression of ligand-receptor pairs in the respective cell types. We performed this analysis on all cell annotations (Supplementary Fig. 22) and focused on interactions between neutrophils and all other cell types that are enriched in *qMCP⁻/⁻* vs. *WT* mice (Fig. 5D). We noted that the neutrophils from tumors generated in *qMCP⁻/⁻; Ntv-a* mice appeared to be enriched in interactions with many tumor clusters (T0, and T2 to T5) through secretion of tumor necrosis factor α (TNF-α) and signaling via TNF-α receptor-I (TNFR-I: p55) and DAG1 on tumor cells (Fig. 5D).

A similar interaction of TNF-α/TNFR-I/II was apparent between neutrophil and endothelial cells (Fig. 5D), which has been shown to be essential for neutrophil transmigration through the endothelial layer[61],

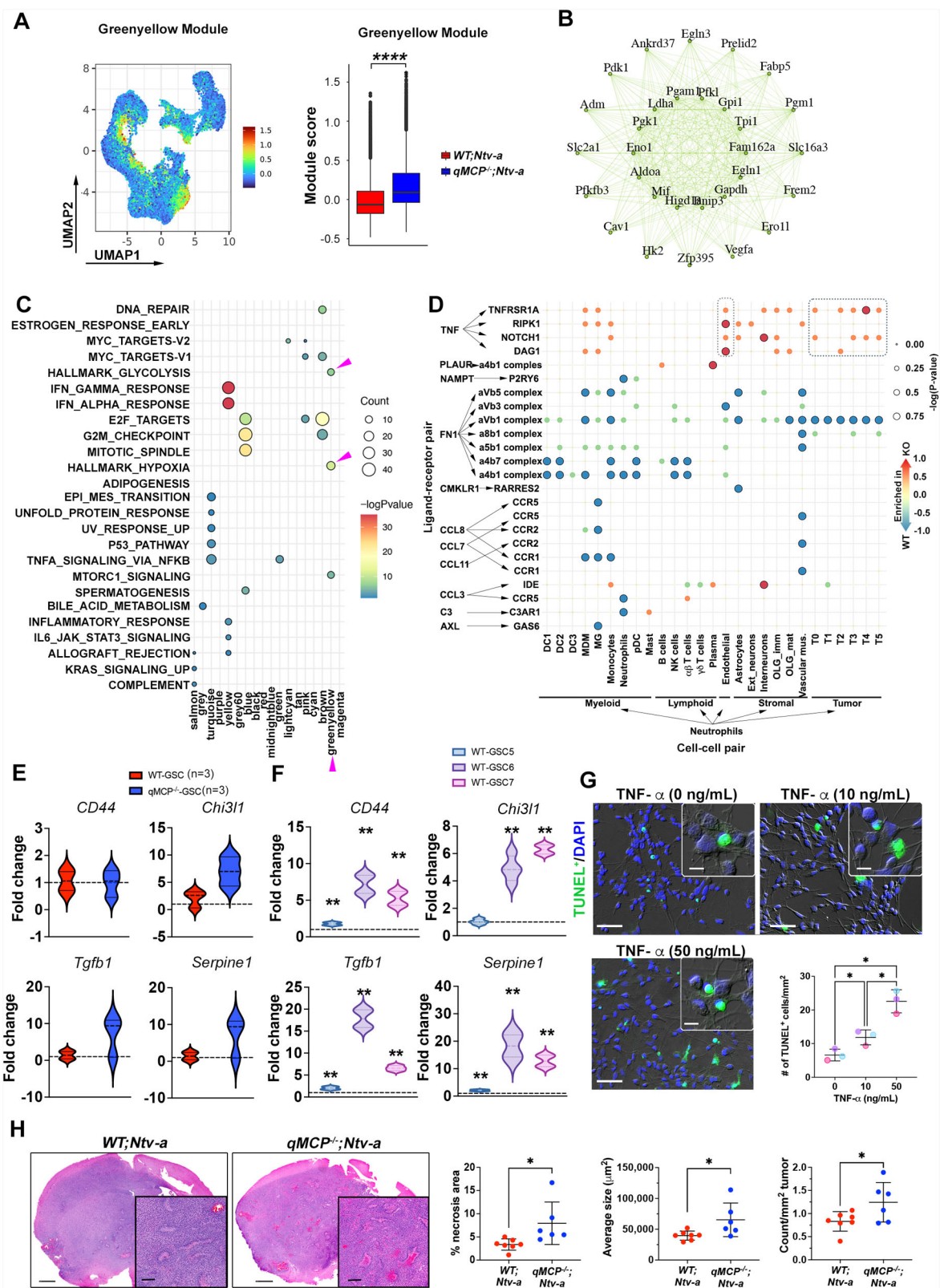

and could potentially explain their increased presence in tumors from *qMCP−/−;Ntv-a* mice.

To confirm neutrophil production of TNF-α, we evaluated *Tnfa* expression across cell types and genotypes examined by scRNA-seq (Supplementary Fig. 23A). Neutrophil, Mg, and MDM have the highest percent expression of *Tnfa,* followed by Mg and to a lesser extent by MDM. The importance of *Tnfa* production by neutrophils in qMCP-

deficient tumors is further bolstered by the decreased proportion of monocytes/MDM in these samples and the fact that Mg are mainly found in peritumoral areas. To examine TNF-α spatial distribution within the tumor, we performed fluorescence in situ hybridization using RNAscope probes against *Tnfa,* along with immunofluorescence staining of IBA1 and Elane. We found mRNA of *Tnfa* was mostly detectable in Elane+ neutrophils (Supplementary Fig. 23B). IVYGap

**Fig. 5 | Neutrophils promote tumor progression by inducing a hypoxic response and necrosis. A** UMAP dimensionality reduction of the "Greenyellow" module score identified by scWGCNA analysis (left). Distribution of the average "Greenyellow" module score in malignant cells (right). Two-tailed Wilcoxon signed-rank test was used on the cell distributions. No adjustment was made. **B** Network graph of the top 30 co-expressed genes in the Greenyellow module. **C** Hallmark pathway gene set enrichment analysis of each WGCNA module. Dot colors represent -log(P-value) and dot sizes represent the number of genes in each Hallmark pathway. Arrowheads indicate biological functions related to the Greenyellow module. Benjamini–Hochberg adjusted (BH-adjusted) hypergeometric test was used. **D** CellphoneDB dotplot showing differentially enriched interactions between ligands (expressed by neutrophil) and receptors (expressed by recipient cells). Dot colors represent the proportion of *WT; Ntv-a* vs. *qMCP⁻/⁻; Ntv-a* enrichment and dot sizes represent the −log(P value) of the differential enrichment. **E** Quantitative real-time PCR for expression of signature MES genes in primary *PDGFB*-driven GBM tumor cell cultures derived from *WT; Ntv-a (WT-GSC)* and *qMCP⁻/⁻; Ntv-a* (qMCP⁻/⁻-GSC)

mice (*N* = 3 independent primary cultures for each genotype), and **F** in *WT-GSCs* treated with 10 ng/mL of TNF-α for 48 hrs. *P* = 0.0030, 0.0001, 0.0002, respectively for *CD44*; *P* = 0.0002 and 0.0000, respectively, for *Chi3l1*; *P* = 0.0010, 0.0000, and 0.0000, respectively for *Tgfb1*; *P* = 0.0003 and 0.0000, respectively for *Serpine1*, where asterisks are present. *N* = 3 independent primary cultures. **G** Representative images of TUNEL staining and quantification of TUNEL positive cells in *WT-GSCs* treated with 10 ng/mL and 50 ng/mL TNF-α for 48 h. *N* = 3 independent primary cultures. Data are presented as mean +/− SD. *P* = 0.0255 (0 vs. 10), 0.0213 (0 vs. 50), and 0.0202 (10 vs. 50). **H** Representative images and quantification of H&E staining for necrosis. *N* = 7 (*WT; Ntv-a*) and 6 (*qMCP⁻/⁻; Ntv-a*) mice. Data are presented as mean +/− SD. *P* = 0.0263, 0.0353, and 0.0448, respectively. Fisher's exact test for D, One-way ANOVA followed by Tucky's post hoc analysis for (**F, G**). Two-sided Student's *t* test for (**E, H**). *P < 0.05, **P < 0.01. Scale bar in (**G**) = 50 μm, scale bar in inset = 20 μm; in (**H**) = 1 mm, in inset = 250 μm. GSC = glioblastoma stem cell. (GSC-5, GSC-6, GSC-7 are primary lines derived from three different tumor-bearing mice). Source data are provided as a Source Data file.

data analysis further corroborated Elane and TNF-related molecules are predominantly transcribed in the peri-necrotic regions (Supplementary Fig. 23C).

To determine whether loss of MCPs in tumor cells drives mesenchymal transition, we generated primary GSC cultures of *PDGFB*-driven GBM generated in *qMCP⁻/⁻; Ntv-a* mice ("qMCP⁻/⁻-GSCs") and compared MES signature gene expression to WT-GSCs by qPCR. There were no differences in expression levels of MES signature genes in qMCP⁻/⁻-GSCs relative to WT-GSCs (Fig. 5E). This result compelled us to hypothesize that PN- > MES transition is not tumor cell autonomous, but rather is a consequence of TME-derived TNF-α stimulation. To this end, we cultured WT-GSCs in the absence or presence of TNF-α at 10 ng/mL, a concentration previously demonstrated to promote tumor growth in human GSCs[54]. Quantitative PCR of *Cd44, Chi3l1, Tgfb1*, and *Serpine1* showed that all these signature MES molecules were increased in these cells following TNF-α treatment, although to various levels (Fig. 5F). We also observed similar results with MCP⁻/⁻-GSCs grown in TNF-α (Supplementary Fig. 24). To determine whether TNF-α increases glycolysis of qMCP⁻/⁻-GSCs in vitro, we grew these cells in the absence or presence of TNF-α at 10 ng/mL under both normoxic and hypoxic conditions. We did not observe any differences in glycolysis (Supplementary Fig. 25). Since the scRNA-seq results demonstrated that neutrophil recruitment chemokines are expressed in comparable amounts by both non-malignant (mainly neutrophils and ECs) and malignant cells (Supplementary Fig. 17), we wanted to determine whether TNF-α can contribute to neutrophil recruiting chemokine expression by tumor cells. We stimulated WT-GSCs with 10 ng/mL TNF-α for 48 hrs and performed qPCR for *Cxcl1, Cxcl2, Cxcl3*, and *Cxcl5*. We show that TNF-α increases the expression of neutrophil recruitment chemokines by tumor cells (Supplementary Fig. 26), which can further contribute to the neutrophil influx in qMCP-deficient tumors. These results support the existence of a pro-tumorigenic loop by neutrophils and tumor cells in qMCP-deficient tumors, such that neutrophils infiltrate the tumors, express their own recruitment chemokines, and via TNF-α production, signal to tumor cells in a paracrine manner to induce expression of recruitment chemokines to further increase their infiltration.

To dissect tumor-promoting functions of TNF-α, we performed an in vitro proliferation/viability assay using primary WT-GSC cultures (Supplementary Fig. 27). At a low concentration (10 ng/mL) TNFα promotes tumor growth, while at a higher concentration (>50 ng/mL), TNF-α reduced the viability of the cells (Supplementary Fig. 27A). Increased viability could be a consequence of increased proliferation. To this end, we performed cell cycle analysis using flow cytometry. We found that cells treated with TNF-α (10 ng/mL) were increased in the G2/M phase, while decreased in the G0/G1 phase (Supplementary Fig. 27B). Prior work has shown that TNF-α induces human glioma cell

death[62] consistent with our observation that increased necrotic regions exist in tumors from *qMCP⁻/⁻; Ntv-a* mice. We performed a TUNEL assay to examine cell death in the presence of TNF-α. Even at a lower concentration (10 ng/mL), TNF-α induced tumor cell death; however, cell death was increased further at a higher TNF-α concentration (50 ng/mL, Fig. 5G).

Along with the observations that neutrophils aggregate around necrosis in the tumor tissues (Fig. 2G), these results suggest that neutrophils contribute to tumor progression by facilitating necrosis formation and facilitating PN- > MES transition in vivo. In agreement, we observed increased necrotic areas in tumors from *qMCP⁻/⁻; Ntv-a* mice (Fig. 5H). In addition, the average size and total occurrence of the necrotic cores were also increased in *qMCP⁻/⁻;Ntv-a* mice (Fig. 5H).

## Genetic *Cxcl1* loss extends the survival of *qMCP⁻/⁻;Ntv-a* mice

In light of the prominent neutrophil infiltration seen in *qMCP⁻/⁻; Ntv-a* mice, we hypothesized that genetic deletion of *Cxcl1* might reduce neutrophil infiltration and extend survival[63]. To this end, we first generated GBM in *WT; Ntv-a* and *Cxcl1⁻/⁻; Ntv-a* mice by co-injecting RCAS-*shp53* and RCAS-*PDGFB* (Fig. 6A, top). However, Kaplan–Meier curves demonstrated no survival differences between these two genotypes (Fig. 6A, bottom). When we analyzed the myeloid compartment of the TME by spectral flow cytometry (Fig. 6B), no significant changes in infiltrating myeloid cells were observed, although decreased Mg and neutrophil abundance were noted (Fig. 6C). Since Mg and neutrophils account for only a small portion of the myeloid cells in *PDGFB*-driven mGBM, it is not surprising that further reduction of either had no impact on the survival of GBM-bearing mice.

Given that neutrophil influx and *Cxcl1* expression were increased in tumors generated in *qMCP⁻/⁻; Ntv-a* mice, we reasoned that genetic deletion of *Cxcl1* in *qMCP⁻/⁻; Ntv-a* mice might reverse these phenotypes and prolong the survival of tumor-bearing mice. De novo *PDGFB*-driven tumors were thus generated in *qMCP⁻/⁻; Cxcl1⁻/⁻; Ntv-a* mice, resulting in extended survival (Fig. 6D). Spectral flow cytometry analysis of the myeloid cells (Fig. 6E) revealed a reduction in total BMDM infiltrates (Fig. 6F), but Ly6c^Hi inflammatory monocytes remained low, comparable to that seen in *qMCP⁻/⁻;Ntv-a* mice (Fig. 6F). In addition, the infiltrating neutrophils were reduced by *Cxcl1* deletion to less than 50% of that seen in *qMCP⁻/⁻; Ntv-a* mice (Fig. 6F), establishing a critical role for *Cxcl1* in recruiting neutrophil GBM chemoattraction. Next, we examined neutrophil recruitment chemokines by qPCR (Fig. 6G) and observed a reduction in *Cxcl2* and −*3* expression, but a compensatory increase in *Cxcl5* (Fig. 6G). It is interesting to note that neutrophil recruitment chemokines are localized adjacent to each other on chromosome 5, reminiscent of the MCPs, suggesting a similar compensatory mechanism might also exist.

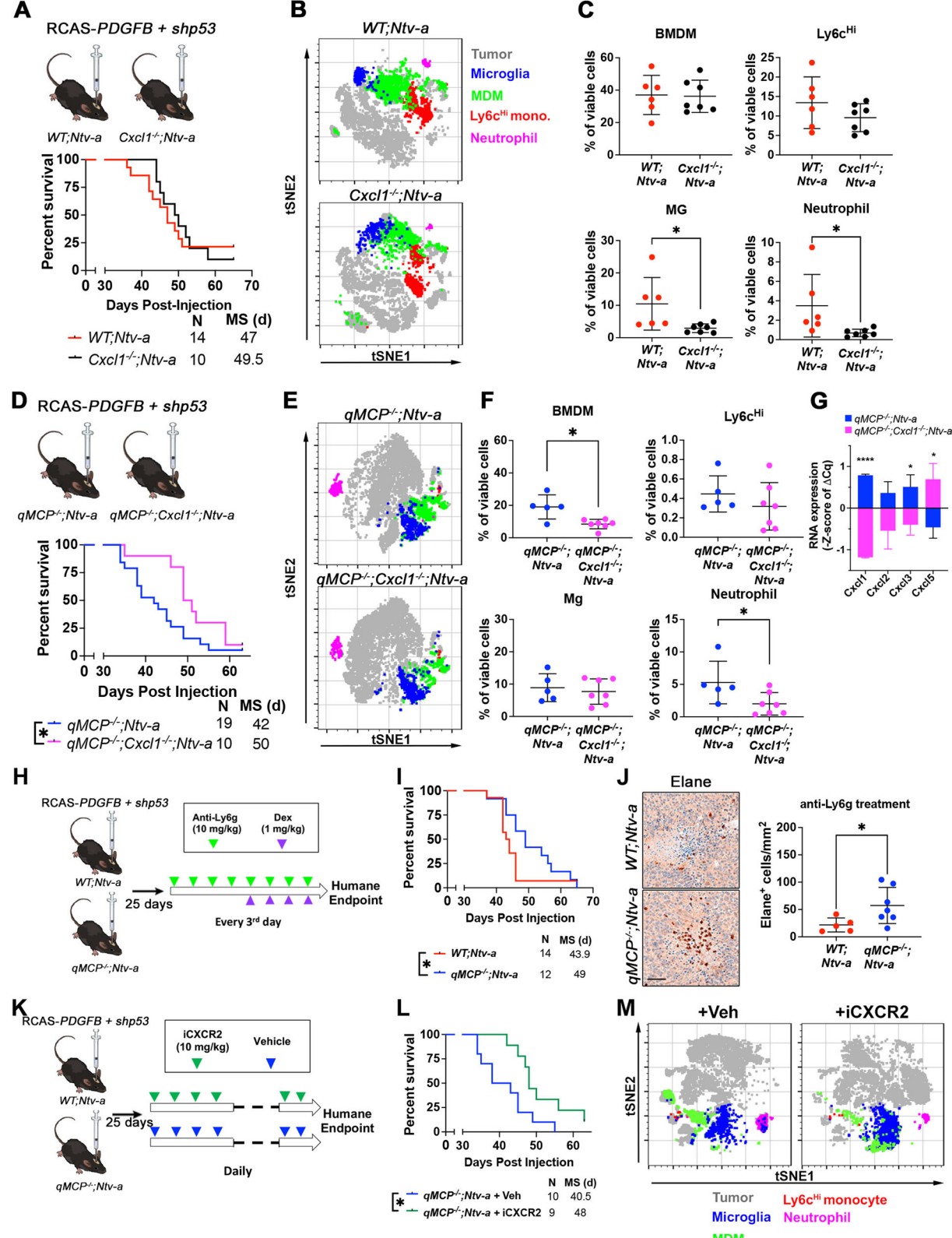

**Pharmacological targeting of neutrophils prolongs the survival of *qMCP⁻/⁻;Ntv-a* mice**

In light of the finding that *Cxcl1* deletion reduces GBM neutrophil infiltration and extends survival, we sought to determine whether pharmacological inhibition of neutrophils would produce the same effects. For these experiments, we employed two strategies to reduce neutrophils. First, we adopted the widely used anti-Ly6g antibody

neutrophil depletion paradigm. Starting day 25 after tumor initiation until the humane endpoint, we injected anti-Ly6g antibodies (IP, 200 μg/mouse, once every third day) to deplete neutrophils from the circulation and from the tumors (Fig. 6H). Following anti-Ly6g treatment, the survival time of tumor-bearing *qMCP⁻/⁻; Ntv-a* mice was prolonged relative to *WT; Ntv-a* controls (Fig. 6I). Surprisingly, when we quantified Elane⁺ neutrophils at the endpoint of the survival

**Fig. 6 | Genetic deletion of *Cxcl1* or pharmacological inhibition of neutrophil recruitment extends the survival of *qMCP⁻/⁻;Ntv-a* mice. A** Schematic illustration and Kaplan−Meier survival curves of *PDGFB*-driven tumors generated in *WT; Ntv-a* and *Cxcl1⁻/⁻; Ntv-a* mice (*Cxcl1* is lost in both tumor cells and TME). **B** tSNE plots illustrating myeloid composition in tumors. **C** FACS quantification of myeloid subtypes. Each dot represents an independent mouse. Data are presented as mean + /− SD. *P* = 0.0330 and 0.0420, respectively, where asterisks are present. *N* = 6 (*WT; Ntv-a*) and 7 (*Cxcl1⁻/⁻; Ntv-a*) mice. **D** Schematic illustration and Kaplan−Meier survival curves of *PDGFB*-driven tumors generated in *qMCP⁻/⁻; Ntv-a* and *qMCP⁻/⁻; Cxcl1⁻/⁻; Ntv-a* mice. *P* = 0.0389 by Log-rank test and *P* = 0.0167 by Gehan−Breslow−Wilcoxon (GBW) test. **E** tSNE plots illustrating myeloid composition in tumors. **F** FACS quantification of myeloid subtypes. *N* = 5 (*qMCP⁻/⁻; Ntv-a*) and 7 (*qMCP⁻/⁻; Cxcl1⁻/⁻;Ntv-a*) mice. Data are presented as mean + /− SD. *P* = 0.0061 and 0.0470 where asterisks are present. **G** Real-time qPCR on tumors from (**F**) at endpoint of survival. Centerline shows Z score equals to 0, whiskers indicate standard deviation. *N* = 9 (independent *qMCP⁻/⁻; Ntv-a* mice) and 6 (independent *qMCP⁻/⁻; Cxcl1⁻/⁻; Ntv-a* mice). Data are presented as mean + /− SD. *P* = 0.0000, 0.0453, and 0.0218, respectively, where asterisks are present. **H** Schematic illustration of treatment paradigm using anti-Ly6g antibodies. **I** Kaplan−Meier survival curves of *WT; Ntv-a* and *qMCP⁻/⁻; Ntv-a* mice treated with anti-Ly6g antibodies. *P* = 0.0260 by Gehan−Breslow−Wilcoxon (GBW) test and *P* = 0.1319 by Log-rank test. **J** Representative images and quantification of Elane⁺ neutrophils in tumors from mice at the endpoint of survival. *N* = 5 (*WT; Ntv-a*) and 7 (*qMCP⁻/⁻; Ntv-a*) mice. Data are presented as mean + /− SD. **K** Schematic illustration of treatment paradigm using CXCR2 antagonist. **L** Kaplan−Meier survival curves of *qMCP⁻/⁻; Ntv-a* mice treated with or without iCXCR2. *P* = 0.0158 by Log-rank test and *P* = 0.0122 by Gehan−Breslow−Wilcoxon (GBW) test. **M** tSNE plots illustrating myeloid composition in tumors. Two-sided Student's *t* test for (**C**, **F**, **G**, **J**) and Gehan−Breslow−Wilcoxon (GBW) test and Log-rank test for (**D**, **I**, **L**). \**P* < 0.05. BMDM bone marrow-derived myeloid cells, Mg microglia, iCXCR2 CXCR2 inhibitor, Dex dexamethasone, MS median survival. Veh vehicle. Scale bar = 50 μm. Source data are provided as a Source Data file.

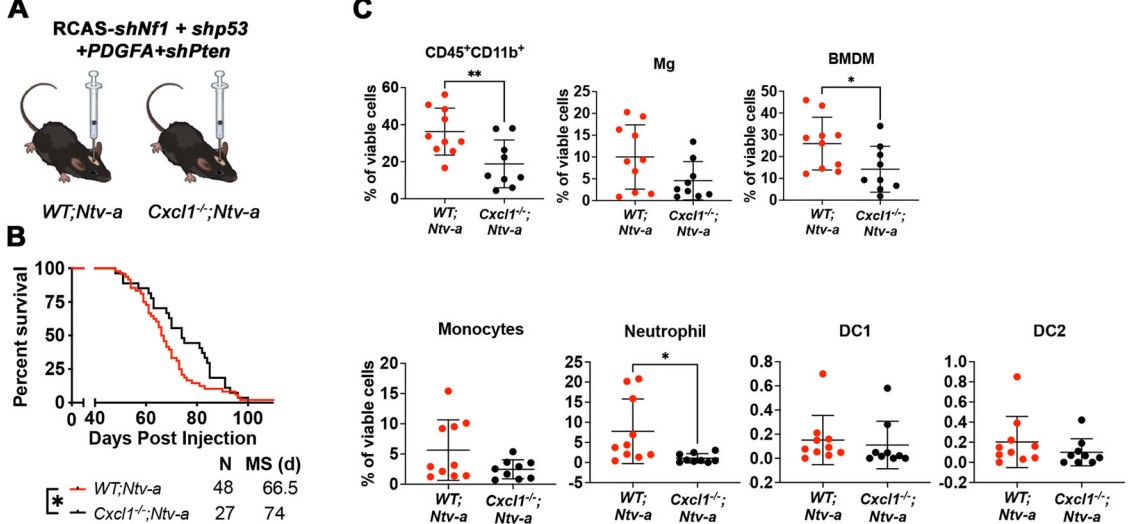

**Fig. 7 | Genetic loss of *Cxcl1* results in decreased neutrophil infiltration and extended survival of *Nf1*-silenced tumor-bearing mice. A** Schematic Illustration of generation of *Nf1*-silenced tumors using *WT; Ntv-a* and *Cxcl1⁻/⁻; Ntv-a* mice and **B** corresponding Kaplan−Meier survival curves. *P* = 0.0479 by GBW test and *P* = 0.0906 by Log-rank test. **C** Quantification dot plots of various myeloid subsets by spectral flow cytometry. Each dot represents an independent mouse. Data are presented as mean + /− SD. *P* = 0.0085, 0.0386, and 0.0241, respectively where asterisks are present. *N* = 10 (*WT; Ntv-a*) and 9 (*Cxcl1⁻/⁻; Ntv-a*) mice. Log-rank and GBW test for (**B**) and two-sided Student's *t* test for (**C**). \**P* < 0.05, \*\**P* < 0.01. Source data are provided as a Source Data file.

experiment, neutrophils remained elevated in *qMCP⁻/⁻; Ntv-a* mice (Fig. 6J). As recently documented, anti-Ly6g neutrophil depletion effects are transient, occurring only at the initial stage of treatment, followed by an effective rebound[48,64]. To better understand the dynamics of neutrophil response to Ly6g depletion, we collected blood from naïve animals treated with anti-Ly6g antibody at various time points and analyzed blood neutrophil counts using a Cytospin assay (Supplementary Fig. 28A). In keeping with a prior report[48], our results showed a transient depletion and a gradual rebound of neutrophils in the blood (Supplementary Fig. 28B), accounting for our findings in the tumor tissues (Fig. 6J). It is important to note that, independent of genotype, mice treated with anti-Ly6g antibody exhibit treatment-induced seizures after 4 doses, which we attributed to increased systemic inflammation. To counteract this adverse effect, we administered dexamethasone at 1 mg/kg (IP, every third day) (Fig. 6H).

Second, we inhibited neutrophil recruitment using a CXCR2 antagonist—a potent and selective small molecule inhibitor (iCXCR2) – SB225002[65]. CXCR2 is the only functional receptor for Cxcl1, −2, −5, and −15 in mice, where it is crucial for neutrophil recruitment and the regulation of vascular permeability[66–68]. The potency of SB225002 in inhibiting neutrophil chemotaxis in vitro[65] and in vivo[66,69] had been demonstrated in various disease contexts, including cancer[70]. GBM-

bearing *qMCP⁻/⁻; Ntv-a* mice were treated with either vehicle or iCXCR2 25 days after tumor initiation (Fig. 6K), resulting in increased survival (Fig. 6L). We performed a comprehensive analysis of myeloid composition by spectral flow cytometry (Fig. 6M). Similar to anti-Ly6g treatment, no difference in neutrophil numbers was observed in iCXCR2-treated tumors at the terminal stage of cancer (Fig. 6M). Collectively, these results demonstrate that strategies aiming at reducing neutrophils in *qMCP⁻/⁻; Ntv-a* mice only transiently prolong survival as a result of rebound neutrophil infiltration.

### Decreased neutrophil, but not monocyte, infiltration prolongs the survival of *Nf1*-silenced GBM-bearing mice

Our results show that genetic reduction of neutrophil infiltration only extends the survival of *qMCP⁻/⁻; Ntv-a* mice with *PDGFB*-driven tumors that undergo PN- > MES transition, but not in *WT;Ntv-a* mice. To determine whether this results from increased neutrophils in tumors with a MES signature, we generated *Nf1*-silenced murine GBM using the RCAS/tv-a system. Based on the expression profile, we have previously demonstrated that *Nf1*-silenced murine GBM closely resembles human MES subtype, where the majority of NF1 del/mut human GBM samples are found[4,7,58]. We generated *Nf1*-silenced GBM in *WT;Ntv-a* and *Cxcl1⁻/⁻; Ntv-a* mice and performed Kaplan−Meier survival analysis (Fig. 7A, B).

Loss of *Cxcl1* extended the survival of *Nf1*-silenced GBM-bearing mice relative to their *WT* counterparts. Next, we used multi-parameter spectral flow cytometry to analyze the composition of myeloid cells in these tumors (Fig. 7C). Based on the combination of multiple surface markers (gating strategy shown in Supplementary Fig. 12), we identified that the total myeloid cells, comprised of both brain-resident Mg and infiltrating BMDM, are reduced in *Cxcl1*-deficient tumors compared to their *WT* counterparts. While no significant changes were observed in resident Mg and monocytes, there was a reduction in neutrophils (Fig. 7C). Quantitatively, we did not observe any difference in DC1 and DC2 (Fig. 7C), and in any other lymphoid populations (Supplementary Fig. 29).

To answer the question whether loss of MCPs would further increase neutrophil influx in *Nf1*-silenced GBM, we generated *Nf1*-silenced GBM in *WT; Ntv-a* and *qMCP⁻/⁻; Ntv-a* mice and performed survival analyses. Loss of MCPs had no effect on the survival of tumor-bearing mice (Supplementary Fig 30A, the *WT; Ntv-a* group was also serves as control for Fig. 8B, since the injections were performed at the same time). Histological evaluation of qMCP-deficient and WT tumors showed reduction of tumor-associated macrophages by IBA1 staining (Supplementary Fig. 30B) with no changes in P2RY12, CD31, OLIG2, GFAP, and CD44 positive areas (Supplementary Fig. 30C–G). There were no changes in Elane⁺ neutrophil numbers (Supplementary Fig. 30H), suggesting that decreased monocyte influx was not associated with increased neutrophil recruitment in *Nf1*-silenced GBM. Nevertheless, we asked whether genetically ablating *Cxcl1* in *qMCP⁻/⁻;Ntv-a* mice can extend the survival of *Nf1*-tumor-bearing mice. Indeed, we observed a significant prolonged survival of *Nf1*-silenced GBM in *qMCP⁻/⁻; Cxcl1⁻/⁻; Ntv-a* mice compared to *qMCP⁻/⁻;Ntv-a* mice, similar what we have shown with *Nf1*-silenced tumors in *Cxcl1⁻/⁻; Ntv-a mice* (Supplementary Fig. 31).

### Abrogation of neutrophil, but not monocyte, infiltration increases the survival of HCC-bearing mice

Our results using monocyte^Hi^neutrophil^Lo^ *PDGFB*-driven GBM models showed that abolishing monocyte influx resulted in compensatory recruitment of neutrophils, while in monocyte^Int^neutrophil^Int^ *Nf1*-silenced GBM, abolishing monocyte recruitment did not. These results compelled us to hypothesize that compensation between monocyte and neutrophil takes place when either targeted population is the predominant infiltrator in the tumors. To test this hypothesis and to determine whether this is also a CNS- and/or GBM-specific phenomena, we employed a GEMM model of hepatocellular carcinoma (HCC). In contrast to *PDGFB*-driven GBM, and more so than *Nf1*-silenced GBM, HCC tumors are mainly populated by neutrophils, mirroring the monocyte:neutrophil ratio (~1:3) seen in the blood of *WT* animals (Fig. 1H). Two of the most frequently altered genes in human HCC are *MYC* (amplified in 17% of HCCs) and *TP53* (deleted or mutated in 33% of HCCs)[71], where they tend to co-occur[71,72]. For this reason, we generated murine HCCs with *MYC* overexpression and *TP53* loss by hydrodynamic tail-vein injections of a transposon vector co-expressing MYC and luciferase (*MYC-Luc*) and a CRISPR vector expressing a sgRNA targeting *p53* (*sg-p53*)[72,73] in *qMCP⁻/⁻* (abrogated monocyte infiltration), *Cxcl1⁻/⁻* (decreased neutrophil infiltration), and *WT* C57BL/6 mice (Fig. 8). Because of the known difference in median survival between male and female mice in this HCC model, we stratified the mice by gender and analyzed each sex separately. Liver luciferase expression measured by bioluminescence imaging (BLI) at day 7 demonstrates equal intensity between all genotypes, revealing similar in vivo transfection efficacy of the plasmids (Fig. 8A, B). No differences in BLI signal were observed 21 days post-injection (Fig. 8A, B). While Kaplan–Meier survival analysis demonstrated that deleting all *MCP* genes had no effect on survival time of HCC mice, knocking out *Cxcl1* extended survival, regardless of gender (Fig. 8C). Because of similar outcomes

between males and females, subsequent studies were performed only using male mice. No significant differences in total myeloid and lymphoid cell populations were observed between genotypes (Fig. 8D). However, when we specifically evaluated the myeloid compartment of HCC by spectral flow cytometry (Fig. 8E, gating strategies in Supplementary Fig. 32), there was a reduction of monocytes and an increase in neutrophil infiltration in *qMCP⁻/⁻* tumors (Fig. 8F). These results suggest that similar to *PDGFB*-driven-GBM, abolishing monocyte recruitment leads to compensatory neutrophil infiltration in HCC but confers no effect on survival. We also observed decreased neutrophil infiltration which was associated with a compensatory increase in monocyte infiltration in *Cxcl1*-deficient tumors (Fig. 8F) and a decreased neutrophil-to-monocyte ratio (Fig. 8G). Whereas no difference in Kupffer cells (KCs) was observed (Fig. 8H), *Cxcl1* deletion resulted in increased liver capsular macrophages (LCMs) (Fig. 8H), consistent with prior reports demonstrating that LCMs are replenished from blood monocytes, while KCs are embryonically derived and capable of self-renewal in situ[74]. In-depth analysis of the lymphoid compartment showed no changes between the three genotypes (Supplementary Fig. 33).

Taken together, these observations establish that blocking monocyte chemoattraction results in increased neutrophil infiltration in *PDGFB*-driven tumors. Increased neutrophil recruitment induces TNF-a-dependent PN to MES transition in *PDGFB*-driven GBM. Inhibiting neutrophil infiltration in monocyte-deficient *PDGFB*-driven-bearing mice improves survival. Decreased neutrophil recruitment but not blocking monocyte recruitment prolongs the survival of *Nf1*-silenced tumors. Blocking neutrophil, but not monocyte infiltration prolongs survival of HCC-bearing mice (Fig. 8I).

## Discussion

Despite intensive efforts over the last two decades, modulating tumor-associated myeloid cells to treat solid tumors, including GBM, has proven exceptionally difficult. This is in large part due to an incomplete understanding of the myeloid heterogeneity during tumor progression and treatment. A classic example is CSF1R antagonism, where its variable therapeutic efficacy is heavily impacted by tumor type and/or models studied and has been shown to be largely ineffective in established tumors[52,75–77]. These studies suggest that myeloid cells evolve and attain independence from CSF1R inhibition as diseases progress, thereby become elusive to therapy. To gain insights into the complexity of immune cell responses following myelo-inhibition, we utilized numerous complementary approaches in this study, leveraging unique chemokine knockout GEMMs, scRNA-seq, human transcriptomics and proteomics data, and two different murine models of GBM with differential myeloid recruitment and HCC.

First, we demonstrate that genetic loss of individual MCP genes in stromal cells results in improved survival of *PDGFB*-driven tumor-bearing mice with no change in myeloid content, while loss of individual MCPs from both stromal *and* tumor cells abolishes survival advantage. These results suggest a redundancy in function of MCP members and compensatory changes in monocyte populations following their elimination. It also motivated us to create a mouse model that is deficient of all MCP members. As expected, depletion of all MCPs from both the stroma and the tumor compartments abolishes monocyte recruitment by trapping them in BM. To our surprise a neutrophil influx and a concomitant rise in neutrophil chemotactic cytokines *Cxcl1* and *Cxcl2* accompanied monocyte depletion (Fig. 2). In contrast to monocytes, due to their lower abundance in GBM, the role of neutrophils has remained less known[78,79]. Recent studies have demonstrated that neutrophil infiltration and activation are markers of poor glioma prognosis[80]. Peripheral cellular immunosuppression in patients with GBM was shown to be associated with neutrophil degranulation and elevated levels of circulating Argl[81]. Increased recruitment of neutrophils during anti-VEGF therapy was shown to

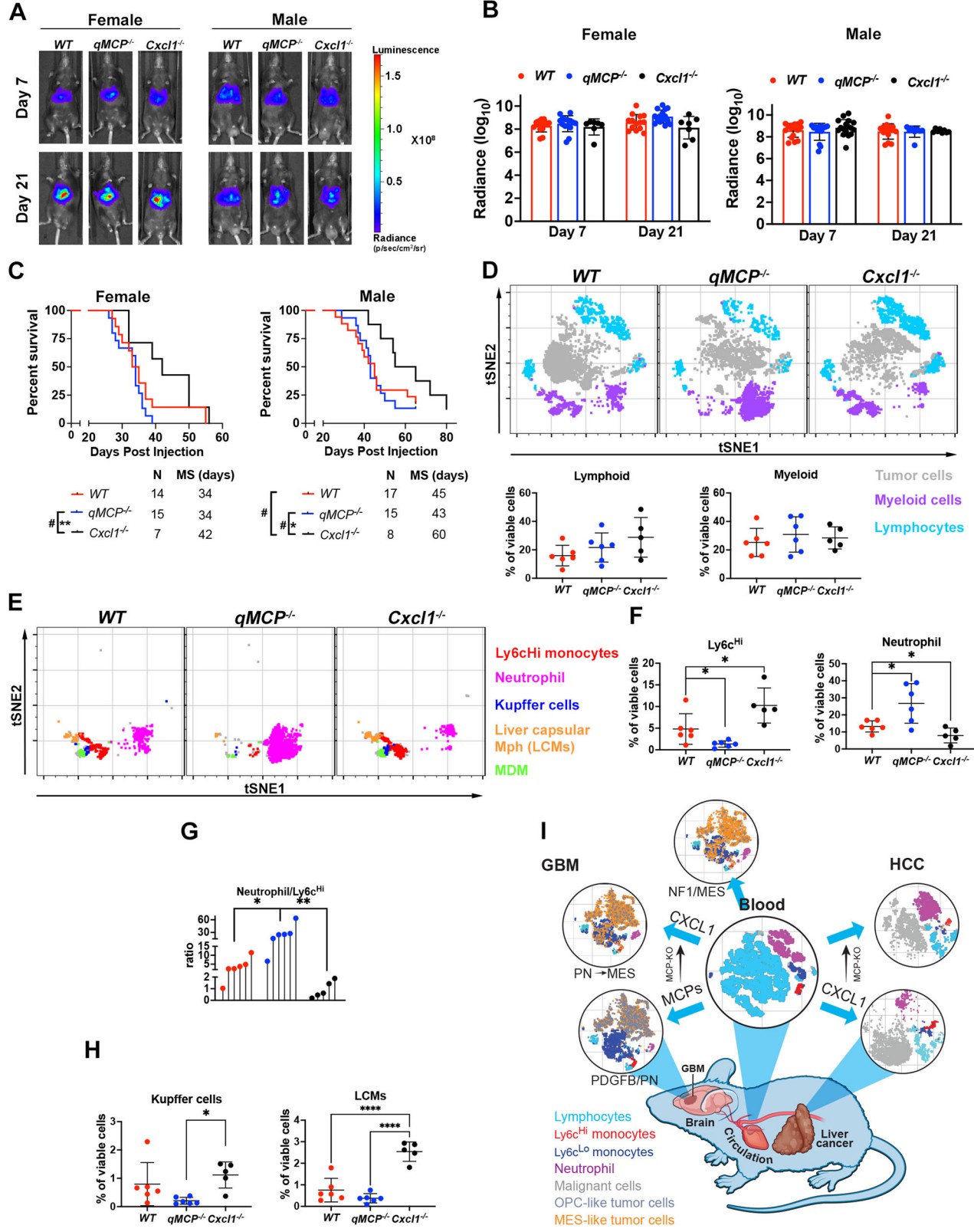

promote treatment resistance via S100A4[82]. Together, these observations elucidate the lack of difference in survival duration between *WT* and *qMCP*-deficient mice in this study.

Functional investigation of neutrophil in GBM is a nascent field[79]. It has been shown at the onset of GBM formation, neutrophils have an anti-tumoral effect, but adopt a tumor-supportive phenotype as tumor progression occurs[16]. One of the mechanisms by which neutrophils

exert their pro-tumorigenic function in GBM is via induction of ferroptosis and tumor necrosis[48]. Similarly, spatial analysis of tumor-associated neutrophil by IHC in this study reveals that the majority of these cells gather in or around necrotic areas (Fig. 2). Inference of ligand-receptor pairs from scRNA-seq data suggest that neutrophils release TNFα, which in a paracrine fashion induces expression of MES signature genes in WT-GSC cultures, increases tumor necrosis and a

**Fig. 8 | Decreasing neutrophil but not abolishing monocyte recruitment decreases tumor growth and extends the survival of HCC-bearing mice.**
**A** Representative images and **B** corresponding quantification graphs of biolumi-nescence imaging at 7 and 21 days after tumor initiation. $N = 12$ (individual female *WT* mice), 13 (individual female *qMCP*$^{-/-}$ mice) and 7 (individual female *Cxcl1*$^{-/-}$ mice). $N = 17$ (individual male *WT* mice), 15 (individual male *qMCP*$^{-/-}$ mice) and 8 (individual male *Cxcl1*$^{-/-}$ mice). Data are presented as mean $+/-$ SD. **C** $P = 0.0051$ by Log-rank test and $P = 0.0281$ by GBW test, respectively, between female *qMCP*$^{-/-}$ and female *Cxcl1*$^{-/-}$ mice. $P = 0.047$ by GBW test between male *WT* and male *Cxcl1*$^{-/-}$ mice; $P = 0.0469$ by Log-rank test and $P = 0.0259$ by GBW test, respectively between male *qMCP*$^{-/-}$ and male *Cxcl1*$^{-/-}$ mice. **D** tSNE plots and flow cytometry quantification of lymphoid and myeloid cells in HCC-bearing mice. $N = 6$ (*WT*), 6 (*qMCP*$^{-/-}$) and 5 (*Cxcl1*$^{-/-}$) mice. Data are presented as mean $+/-$ SD. **E** tSNE plots illustrating myeloid cells examined by spectral flow cytometry. **F** Quantification of monocytes and

neutrophils by spectral flow cytometry. Each dot represents an independent mouse. Data are presented as mean $+/-$ SD. $P = 0.0406$ (WT vs. *qMCP*$^{-/-}$) and 0.0415 (WT vs. *Cxcl1*$^{-/-}$) for Ly6c$^{Hi}$ monocytes; and $P = 0.0206$ (WT vs. *qMCP*$^{-/-}$) and 0.0468 (WT vs. *Cxcl1*$^{-/-}$) for Neutrophils. $N$ is the same as in (**D**). **G** Lollipop plot showing neutrophil to Ly6c$^{Hi}$ monocytes ratio. Each dot represents an independent mouse. $P = 0.0124$ (WT vs. *qMCP*$^{-/-}$) and 0.0067 (*qMCP*$^{-/-}$ vs. *Cxcl1*$^{-/-}$). $N$ is the same as in (**D**). **H** Quantification of Kupffer cells and LCMs by spectral flow cytometry. Data are presented as mean $+/-$ SD. $P = 0.0308$ (*qMCP*$^{-/-}$ vs. *Cxcl1*$^{-/-}$) for Kupffer cells; and $P = 0.0000$ (WT vs. *Cxcl1*$^{-/-}$) and 0.0000 (*qMCP*$^{-/-}$ vs. *Cxcl1*$^{-/-}$) for LCMs. $N$ is the same as in (**D**). One-way ANOVA with Tukey's post hoc test for (**F–H**) and Log-rank and GBW for (**C**). *$P < 0.05$, **$P < 0.01$, ***$P < 0.001$ and #$P < 0.05$ by GBW test. **I** Summary Illustration of findings. MS median survival. $N =$ number of mice. LCM liver capsular macrophages. Source data are provided as a Source Data file.

hypoxic response in vivo, in addition to inducing expression of neu-trophil recruitment chemokines by tumor cells to further facilitate their infiltration (Fig. 5). Future studies should address whether inhi-biting TNF in the TME or ablating TNF receptors on tumor cells will reverse neutrophil-derived resistance to monocyte targeted therapies in tumors with PN signature.

We and others have demonstrated that murine GBM models clo-sely resembling the human MES GBM subtype have increased numbers of neutrophils relative to PN- and CL-like GBM models[4,16]. Our findings that *PDGFB*-driven tumors in *qMCP*-deficient mice have increased expression of MES genes, increased expression of MES marker CD44, and increased hypoxia upon IHC staining are consistent with a transi-tion from a PN->MES GBM subtype (Fig. 4)[14]. To evaluate clinical relevance of our observations, we analyzed NanoString data of *IDH-WT* and *IDH-Mut* tumors for neutrophil and their recruitment chemokines. In agreement, we find an increased presence of neutrophil and neutro-attractant chemokines in human MES subtype samples (Fig. 3). When a larger patient cohort from TCGA was stratified based on their IL8 (*CXCL8*) expression, higher expression is found to be associated with inferior survival. Overall, our results indicate that monocyte:neutrophil ratio can define tumor signature, highlighting the essential roles these two cell types play in shaping tumor cell expression profiles and crafting the evolving TME.

We further confirmed neutrophils have tumorigenic role not only in tumors that show PN->MES shift, but also in primary de novo *Nf1*-silenced murine GBM, which closely clusters with human MES subtype[4,58]. Similar results were documented in *NRas*-[83] and *HRas*-dri-ven GBM models[16]. Although *RAS* is a rare genetic alteration in adult human GBM (accounting for less than 1% of cases (*NRAS*, *HRAS* and *KRAS* combined—we queried 273 GBM patient samples from the TCGA with mutations and CNA data). And since NF1 is negative regulator of RAS, these results together suggest that increased neutrophil recruit-ment in all these models and our *Nf1-silenced* tumors are attributed to RAS. These results are consistent with published results showing that oncogenic RAS, the most common mutation in human cancers (~19% of human cancers harbor a RAS mutation[84]), induces IL8 expression to increase neutrophil recruitment to tumor sites[85]. In line with this finding, in an oncogenic *Kras (G12D)*-driven mouse model of lung cancer and pancreatic ductal adenocarcinoma (PDAC), neutrophils were shown to be a major population of tumor-immune infiltrates[79,86,87].

Abolishing monocyte recruitment in *Nf1*-silenced murine GBM did not promote compensatory recruitment in neutrophils, and we attributed this to the fact that although enriched neutrophils are not the most abundant myeloid cells, in contrast to monocytes in *PDGFB*-driven GBM. We were curious whether decreasing neutrophil infiltra-tion in neutrophil-enriched tumors would lead to increased monocyte infiltration. Therefore, as a proof-of-principle we selected a well-documented neutrophil-enriched tumor outside of the CNS—HCC. It is well established that neutrophil numbers can serve as powerful

predictors of poor outcome in HCC patients, but mechanisms of their infiltration remain elusive[88,89]. By using a genetic mouse model, we demonstrate that similar to human HCC, murine HCC are enriched with neutrophils and their ratio to monocytes (~3:1) mirror that of the blood of healthy WT mice, contrasting to GBM. Abolishing monocytes had no impact on survival of neutrophil-enriched HCC tumor-bearing mice, similar to what we had shown with abolishing neutrophils in monocyte-enriched *PDGFB*-driven GBM. Decreased neutrophil infil-tration resulted in extended survival and was associated with increased monocyte recruitment, suggesting this compensatory mechanism exists both ways, and only takes place when predominant myeloid subset in tumor is targeted.

An intriguing question remains: why are some tumors enriched in monocytes while other neutrophils? Monocytes are a minority in blood circulation, yet they give rise to the dominant infiltrates in GBM; whereas neutrophils, the most abundant in the blood, rarely invade GBM, except for MES tumors in both human and mice, where they still are not the most abundant infiltrates? It is speculated that *Tp53* loss, either alone or in combination with *KRas* or *Pten*, create micro-environments that preferentially favor neutrophil infiltration in various solid tumors, discussed in detail in review[90]. However, *Tp53* mutation occurs at a very high prevalence across many cancer types, ranging from 30 to 47% in brain, liver, lung, skin, ovarian and many other cancers[91]. Therefore, this widespread presence cannot explain the highly diverse TME across all the tumor types. For instance, two GBM and HCC models used in this study involved *Tp53* silencing, however their TMEs are contrasting in terms of monocytes/neutrophil compo-sitions. In addition, CHI3L1 is a MES gene and was previously shown to increase number of TAMs[92] and we do see increased TAMs and increased levels of CHI3L1 expression in our *Nf1*-silenced tumors[4,58]. OPN contributes to increased TAM number and crosstalk with tumor cells[93], and LOX is important in macrophage recruitment and function in the context of PTEN deficiency[94]. Our *PDGFB*-driven models are in the *Pten WT* context, where recruitment mechanisms are different, as we show. We posit that unique combinations of driver mutations in different tumor types, rather than loss of a universal tumor suppressor gene, play a decisive role in this phenomenon. We show that *Nf1*-silenced murine and human GBMs have increased expression of neu-trophil recruitment chemokines compared to *PDGFRA*-amplified hGBM and *PDGFB*-driven mGBM, which favor monocyte infiltration. Therefore, in-depth analysis of tumor samples created by different driver mutations will uncover potential mechanisms each tumor uses to construct their distinctive TMEs.

In conclusion, we demonstrated that when all MCPs were genetically deleted and monocyte recruitment abolished, GBM adapt to mobilize an influx of neutrophils. Similar compensatory effects exist in HCC. These observations explain the failure of current treatment attempts that pursue single chemokines for targeting. It is therefore suggested to develop combinatorial therapies that are simultaneously directed at both monocytes and neutrophils. Effective treatment can

also be confounded by the complexity where a rebound effect often challenges targeting neutrophil influx. A fundamental understanding of the interplay between driver mutations, monocytes, neutrophils, and other TME cell types using state-of-the-art GEMMs, single-cell resolution measurements, and integrative analysis as utilized in our study will be critical to future pharmacological development aiming at creating long-lasting, dual-function compounds.

# Methods

## Research compliances
All mouse experimental procedures were approved by the Institutional Animal Care and Use Committee (IACUC) of Emory University (Protocol #2017-00633) and the Icahn School of Medicine at Mount Sinai (Protocol #2019-00619 and #2014-0229). Archived formalin-fixed, paraffin-embedded (FFPE) human GBM samples and de-identified clinical information were provided by Emory University. All human tissue samples were obtained de-identified through the biorepository at ISMMS, collected under appropriate consent and utilized under approved Institutional Review Board (IRB) protocol HS#18-00983.

## Generating qMCP knockout mice
The qMCP knockout mice were generated at the Mouse Transgenic and Gene Targeting Core at Emory University. A pair of guide RNAs (upstream: CCCTGGCTTACAATAAAAGGCT, and downstream: CAG-CAGGCCAAATGAGGGGAGG) were designed to recognize the 81k base DNA segments flanking the genes between *Ccl2* and *Ccl8* (inclusive) on chromosome 11. The guide RNAs were synthesized and validated by Sigma (Millipore Sigma). The guide RNAs, CRSPR/cas9 mRNA, and a donor repair oligonucleotide (5'-TCACTTATCCAGGGTGATGC-TACTCCTTGGCACCAAGCACCCTGCCTGACTCCACCCCCCAGGTGTT-CAAGGGTTCCTGTGTATTATTTGGGTTTCATTTTATGGGGTTCAAGT-GAAGGA-3') were co-injected into C57BL/6 N (RRID:MG:6198353) zygotes and transferred to surrogate dames. Two founder female mice were identified by PCR, and verified by DNA sequencing. We back-crossed founder #5 to the C57BL/6J strain for over 10 generations and the progeny's genetic background was confirmed to be C57BL/6J via genetic monitoring service provided by Transnetyx. All the mice were viable and fertile. All subsequent genotyping was done at Transnetyx with the probe set named Gm17268-1.

Newly created qMCP-KO mice will be distributed to interested colleagues upon mutually satisfactory materials transfer agreements.

## Mice used in this study
Mice of both sexes (equal distribution) in the age range of 6–12 weeks were used for experiments. Previously described *Ccl2*−/− (Jackson Laboratory, #004434), *Ccl7*−/− (Jackson Laboratory, #017638), *Ccl8/12*−/− (gifted by Dr. Sabina Islam), *Cxcl1*−/−[95], and *Cxcr2*−/− (Jackson Laboratory, #006848) mice were either maintained as single knockout strains, or cross-bred to the *Ntv-a* mice to generate double or triple knockout strains. *Cx3Cr1*$^{GFP/WT}$; *Ccr2*$^{RFP/RFP}$ and *Cx3Cr1*$^{GFP/WT}$; *Ccr2*$^{RFP/RFP}$ mice were generated from heterozygous breeding pairs (Jackson laboratory #005582 and #017586), backcrossed for more than ten generations. All these mice are in a C57BL/6 background. C57BL/6J mice (#000664) at 6 weeks old were purchased from the Jackson labs. All animals were housed in a climate-controlled (18–23 °C and 40–60% humidity), pathogen-free facility with access to food and water *ad libitum* under a 12-hour light/dark cycle.

## RCAS virus propagation to generate de novo GBM
DF-1 cells (ATCC, CRL-12203, RRID:CVCL_0570) were purchased and grown at 39 °C according to the supplier's instructions. Cells were transfected with RCAS-*hPDGFB*-HA, RCAS-*hPDGFA*-myc, RCAS-shRNA-*p53*-RFP, RCAS-shRNA-*Nf1*, RCAS-shRNA-*Pten*-RFP, and RCAS-*Cre* using a Fugene 6 transfection kit (Roche, 11814443001) according to the manufacturer's instructions. DF-1 cells ($4 \times 10^4$) in 1 µl neurobasal

medium were stereotactically delivered with a Hamilton syringe equipped with a 30-gauge needle for tumor generation[58]. To generate *PDGFB*-driven GBM, a cocktail RCAS-*hPDGFB*-HA and RCAS-shRNA-*p53*-RFP was injected to the right-frontal striatum at AP −1.5 mm and right −0.5 mm from bregma; depth −1.5 mm from the dura surface. To generate NF1-silenced tumors, a mixture of RCAS-shRNA-*Nf1*, RCAS-*hPDGFA*-myc, RCAS-shRNA-*p53*-RFP, RCAS-shRNA-*Pten*-RFP were used. The injection site was aimed at the subventricular zone at coordinates AP −0.0 mm and right −0.5 mm from bregma; depth −1.5 mm from the dura surface[4,58,96]. Mice were continually monitored for signs of tumor burden and were sacrificed upon observation of endpoint symptoms, including head tilt, lethargy, seizures, and excessive weight loss. The maximum tumor burden was not exceeded.

## Orthotopic glioma generation
The same procedure was used as described above, except $3 \times 10^4$ of freshly dissociated tumor cells were injected in the right-frontal striatum AP −1.7 mm and right −0.5 mm from bregma; depth −1.5 mm from the dural surface of recipient animals. Two or three donor tumors of either sex were used for obtaining single cells for orthotopic glioma generation in male and female recipient animals.

## Hydrodynamic tail-vein injection to generate HCC
A sterile 0.9% NaCl solution/plasmid mix was prepared containing DNA. We prepared 11.4 µg of pT3-EF1a-MYC-IRES-luciferase (*MYCluc*), 10 µg of px330-sg-p53 (*sg-p53*), and a 4:1 ratio of transposon to SB13 transposase-encoding plasmid dissolved in 2 mL of 0.9% NaCl solution and injected 10% of the weight of each mouse in volume as previously described[72]. Because two independent "hits" are required for tumor formation in C57BL/6 mice[97], only those hepatocytes that receive the three plasmids (transposon-based, transposase, and CRISPR-based) will have the potential to form tumors. Mice were injected with the 0.9% NaCl solution/ plasmid mix into the lateral tail vein with a total volume corresponding to 10% of body weight in 5 to 7 s. Vectors for hydrodynamic delivery were produced using the QIAGEN plasmid PlusMega kit (QIAGEN). Equivalent DNA concentration between different batches of DNA was confirmed to ensure reproducibility among experiments.

## Luciferase detection
In vivo bioluminescence imaging was performed using an IVIS Spectrum system (Caliper LifeSciences, purchased with the support of NCRR S10-RR026561-01) to quantify liver tumor burden. Mice were imaged 5 min after intraperitoneal injection with fresh D-luciferin (150 mg/kg; ThermoFisher Scientific). Luciferase signal was quantified using Living Image software (Caliper LifeSciences, RRID:SCR_014247). The normalized luciferase signal was calculated by subtracting the background signal. Those mice with a luciferase signal a log of magnitude lower than the average signal were excluded from the study.

## NanoString analysis
Human formalin-fixed, paraffin-embedded (FFPE) tissue scrolls were cut in 10-µm sections for RNA extraction using the RNeasy Lipid Tissue Mini Kit (Qiagen #74804) according to the manufacturer's instructions. RNA integrity was confirmed using a bioanalyzer and samples possessing a DV300% greater than 30 were used. In total, 50 ng of RNA was used for NanoString analysis with the pan-cancer pathways immune panel (NanoString, XT-CSO-HIP1-12). All data analysis was processed and normalized using nSolver Analysis Software version 4.0 (NanoString) and GraphPad Prism 9 (GraphPad Software, RRID: SCR_002798).

## Neutrophil morphology analysis by Cytospin
Whole blood was collected with a heparinized capillary tube (Fisher, 22-362-566) via mandibular vein puncture. Overall, 10 µl of whole

blood was suspended in 1 ml RBS lysis buffer in a 1.5-ml Eppendorf tube, which was kept at 37 °C for 1 min. The tube was centrifuged at $450 \times g$ for 4 min at RT and the pellet was resuspended in 100 µl PBS containing 1% BSA. Cytospin slides (ThermoFisher, 5991056) and filter paper (ThermoFisher, 5991022) were assembled according to the manufacturer's instructions. The cell suspension was then transferred to the funnel that was attached to the slides. The assembly was loaded into the Cytospin centrifuge and run at 800 rpm for 5 min. The slides were air-dried at RT for 20 min before they were plunged into ice-cold methanol for 5 min. The slides were then stained with DAPI (Sigma, 1 µg/ml) for 10 min at RT in the dark. Images of the nuclei were taken with a Leica confocal microscope (Leica, SP8) with a 10X objective, with multiple areas of view were acquired for each sample. The number of nuclei with typical neutrophil nuclear morphology (Fig. 7D) was counted.

## Human tissue samples and pathological appraisal
Human FFPE GBM samples, post-mortem brain specimens, and de-identified clinical information were provided by Emory University. Board-certified neuropathologists graded and diagnosed both the human tumor tissues and murine samples according to the 2016 World Health Organization Classification of Tumors of the Central Nervous System[98]. Gene expression profiling to determine transcriptional subtypes was performed using NanoString nCounter Technology using custom-made probes for 152 genes from the original GBM_2 design[15]. Flash-frozen, de-identified GBM samples and adjacent non-malignant tissues acquired during tumor resection were obtained from Mount Sinai. Whole-genome sequencing was performed to determine patients' IDH mutation status and molecular genotype of the tumors.

## TCGA analysis
U133 Microarray data for the GBM (TCGA, provisional) and Glioblastoma Multiforme (TCGA, Firehose Legacy) dataset were downloaded from cBioPortal[32,33] (https://www.cbioportal.org, RRID: SCR_014555) in August 2019 and 2021 and sorted into subtypes based upon a proprietary key. G-CIMP-positive tumors were excluded from analysis. We included 295 patient samples for which covariate information (survival information, age, and gender) was available. Cox Proportional Hazard Models were fitted in R using age and gene expression as continuous covariates, and gender as a binary variable.

## Tissue processing and immunohistochemistry
Archived FFPE human GBM samples and de-identified clinical information were provided by Emory University. Murine FFPE samples were generated as previously described[15]. The specimens were sectioned at 5 µm thickness, slide-mounted, and stored at −80 °C until use. To process mouse tumor tissues, animals were anesthetized with an overdose of ketamine/xylazine mix and transcardially perfused with ice-cold Ringer's solution. Brains were removed and processed according to the different applications. For H&E tumor validation and immunohistochemistry staining, brains were fixed in 10% neutral buffered formalin for 72 h at room temperature (RT), processed in a tissue processor (Leica, TP1050), embedded in paraffin, sectioned at 5 µm with a microtome (Leica), and mounted on superfrost glass slides (ThermoFisher 3039-002). Slides were rehydrated with tap water and dipped in hematoxylin (ThermoFisher, 7231), bluing agent (ThermoFisher, 22-220-106) and eosin (ThermoFisher, M1098442500) for 1 min each with thorough washes with tap water in-between. Slides were dehydrated with series washes in ethanol and Neo-clear (ThermoFisher, M1098435000) before being mounted in Permount medium (ThermoFisher, SP15-100).

All immunohistochemistry staining was performed on a Leica Bond Rx platform (Leica). Primary antibodies (a full list of primary antibodies used in this study is shown in Supplementary Table 3) used in this study include anti-IBA-1 (1:1,500, FUJIFILM Wako, 019-19741,

RRID:AB_839504), anti-CD31 (1:50, Dianova, DIA-310), anti-CD44 (1:100, BD Pharmingen, 550538, RRID:AB_393732), and anti-OLIG2 (1:500, Millipore, AB9610, RRID:AB_570666). Anti-GFAP (1:10,000, CST, 3670, RRID:AB_561049), anti-Elastase (1:400, Bioss, bs6982R or 1:400, Abcam, ab68672), anti-P2Y12 (1: 500, AnaSpec, SQ-ANAB-78839, discontinued). Appropriate secondary antibodies were purchased from Leica or Vectorlab. Digital images of the slides were acquired by using a Nanozoomer 2.0HT whole-slide scanner (Hamamatsu Photonic K.K) and observed offline with NDP.view2 software (Hamamatsu). Image analysis was performed using Fiji (NIH, RRID:SCR_002285).

## Tumor dissociation and primary cell culturing
Tumor dissociation was performed as previously described. Briefly, tumors were dissected from the brain, minced into pieces <1 mm³, and digested with an enzymatic mixture that includes papain (0.94 mg/ml, Worthington, LS003120), EDTA (0.18 mg/ml, Sigma, E6758), cystine (0.18 mg/ml, Sigma, A8199), and DNase (60 µg/ml, Roche, 11284932001) in 2 ml HBSS (Gibco, 14175-095). Tumor tissues were kept at 37 °C for 30 min with occasional agitation. The digestion was terminated with the addition of 2 ml Ovomucoid (0.7 mg/ml, Worthington, LS003086). Following digestion, single cells were pelleted, resuspended in HBSS, and centrifuged at low speed (84 RCF) for 5 min, before passing through a 70-µm cell strainer.

For Glioma stem cell (GSC) cultures, cells were seeded at $5 \times 10^5$ cells/ml and grown in Neurocult mouse neurobasal medium (Stem Cell Technologies, 5700) supplemented with 10 ng/ml hEGF (Lonza, cc-4017FF), 20 ng/ml basic-hFGF (ThermoFisher, PHG0261), 1 mg/ml Heparin (Stem Cell Technologies, 7980), and NSC Proliferation Supplements (Stem Cell Technologies, 5701). Fresh medium was added to the cultures every 48 hours. For primary monolayer cultures, neurospheres were dissociated with Accutase (Sigma-Aldrich, A6964) to generate single cells, which were then grown as adherent monolayers on coverslips ($\Phi = 12$ mm) or well-plates coated with Geltrex (Life Technologies, A14132-01) prepared according to the manufacturer's instructions.

## Tumor and cultured cell RNA isolation and analysis
At humane endpoint, mice were sacrificed with an overdose of ketamine and immediately perfused with ice-cold Ringer's solution (Sigma-Aldrich, 96724-100TAB). The brain was extracted, and a piece of tumor was immediately snap-frozen in liquid nitrogen for storage at −80 °C. Alternatively, cultured cells were harvested from plates using TRIzol (ThermoFisher, 15596026). RNA was isolated from the frozen tumor pieces or cells with the RNeasy Lipid Tissue Mini Kit (Qiagen, 74804) according to the manufacturer's instructions. RNA quantity was assessed with a NanoDrop 2000 spectrometer, while quality was confirmed via electrophoresis of samples in a 1% bleach gel as previously described[99]. RNA was used to generate cDNA with a First Strand Superscript III cDNA synthesis kit (ThermoFisher, 18080051) according to the manufacturer's instructions and with equal amounts of starting RNA. Quantitative PCR was performed with the validated BioRad PCR primers (Supplementary Table 2, except *Cxcl1* and *Cxcl2* whose sequence were obtained from ref. [100]) using SsoAdvanced Universal green Supermix (BioRad, 1725271). Fold changes in gene expression were determined relative to a defined control group using the $2^{-\Delta\Delta Ct}$ method or by z score, with β-actin or HPRT used as housekeeping genes.

## Anti-Ly6g antibody or iCXCR2 treatment
Tumor-bearing mice were randomly assigned to different experimental groups on the first day of treatment. For neutrophil depletion, tumor-bearing mice received intraperitoneal injections of 200 µg of 1A8 (Ly6g depletion, BE0075-1) or 2A3 (control; BE0089, both from Bio X Cell) antibody per mouse starting from day 25 after DF-1 cell injection. Injections were given every day until mice succumb to disease

and were sacrificed at humane endpoints. Mice were monitored for signs of disease progression as described above. To decrease treatment-induced seizures, mice were given 1 mg/kg Dexamethasone (West-ward) every third day starting day 37.

CXCR2 inhibitor SB225002 (iCXCR2) was purchased from Tocris (#2725) and dissolved in DMSO to make 10 mg/ml solution. This solution is diluted 10× with 0.33% Tween80 (v/v) in saline on the day of treatment. Starting on day 20 after tumor initiation, each mouse assigned to the treatment group received an IP injection of iCXCR2 at 10 mg/Kg daily until humane endpoint.

## Olink multiplex proteomic analysis

Flash-frozen human GBM samples and adjacent non-malignant tissues were weighed and ~30 mg of tissues from each sample were transferred to a 1.5-ml Eppendorf tube. T-PER Tissue Protein Extraction Reagent (ThermoFisher, 78510) containing phosphoSTOP and protein inhibitors cocktail (100 mg/ml, Roche 11836153001) was added to the tube at the ratio of 1 ml buffer per 100 mg tissue. The tissues were homogenized on a sonicator till no chunk of tissue visible, for about 30 s. The extractions were kept in cold room for 1 h with rotation. They were centrifuged at $10,000 \times g$ for 5 min at 4 °C and the supernatants were carefully collected. Protein concentration was determined by Bradford assay (BioRad, 5000001) following the manufacturer's instructions. The final concentration of the samples was standardized to 0.5 mg/ml. Samples were shipped to Olink Proteomics (Watertown, MA) on dry ice overnight. The Olink Immuno-oncology panel that analyses 96 immuno-oncology-related human proteins were utilized. Normalized protein expression (NPX) values were generated and reported by Olink, and subsequently analyzed in-house using Prism (Graphpad) or Morpheus (Broad institute) online tool. PCA analysis and graph were performed with MATLAB (Math-Works, RRID:SCR_001622).

## Hematoxylin and eosin staining to identify necrosis in GBM

Mice were sacrificed at humane endpoint with an overdose of ketamine and xylazine and perfused with 10 ml cold Ringer's solution. The brain was carefully extracted and incubated in 10% formalin for 72 h. The brains were dissected through the middle of the tumor and embedded in paraffin. The paraffin block was trimmed and the brains were sectioned on a microtome (Leica) to cut 5-µm sections. The sections were collected and mounted on a slide for automated hematoxylin and eosin staining as described above. The slides were scanned at ×20 magnification with a whole-slide scanner (Hamamatsu). Tumor area in each section was determined in a blinded fashion in NDP.view2.

## Enzyme-linked immunosorbent assay

Whole blood was collected from anesthetized mice via cardiac puncture. Blood Cell lysates for enzyme-linked immunosorbent assay (ELISA) were collected via sonication of cells in lysis buffer supplemented with protease and phosphatase inhibitors. ELISAs were performed for CCL2 (R&D, DY479), CCL7 (Boster Bio, EK0683), CCL8 (R&D, DY790), CCL11 (R&D, MME00), CCL12 (R&D, MCC120) and CCL5 (R&D, DY478-05) on cell lysates and cell supernatants according to the manufacturer's instructions.

## Immunofluorescence and confocal microscopy

Flash-frozen mouse tumor tissues were embedded in O.T.C and cut into 10-µm sections on a Leica Cryostat and mounted on TOMO Adhesion glass sides (VWR, 10748-172). The sections were washed with PBS, permeabilized with 0.1% Triton X-100, and blocked with 10% normal goat serum and 10% normal donkey serum (Jackson ImmunoResearch). The sections were then incubated in primary antibodies overnight at 4 °C. The following antibodies were used at the stated dilutions: rabbit anti-TEME119 (1:500, Abcam, ab209064), mouse anti-IBA1 (1:500, Cleveland Clinic Hybridoma Core), rabbit anti-Elastase

(1:500, Bioss, BS6982R). Secondary antibodies conjugated to Alexa Fluor 594 or Alexa Fluor 647 (Jackson ImmunoResearch) at a dilution of 1:500 in PBS/2% BSA were applied. For nuclear counterstaining, DAPI was used (Sigma). Fluorescence images were taken on a Leica SP8 confocal microscope. Images were analyzed and quantified with FIJI (NIH). For fluorescent in situ hybridization detecting TNF-α mRNA and co-detection with immunofluorescence of IBA1 and Elane, RNAscope probe (311081-C2) were obtained from Advanced Cell Diagnostics (ACD). Briefly, sections were first deparaffinized using xylene and ethanol. Endogenous peroxidase activity was quenched with hydrogen peroxide reagent for 10 min, followed by antigen retrieval for 10 min in boiling buffer (Vector Labs, H-3300). Immunofluorescence was performed using mouse anti-IBA1 (1:500, Cleveland Clinic Hybridoma Core), rabbit anti-Elastase (1:500, Bioss, BS6982R), followed by protease digestion for 30 min at 40 °C. The TNF-α mRNA probes (311081-C2) were then hybridized for 2 h at 40 °C in a humidity-controlled oven (HybEZ II, ACD). Successive addition of amplifiers was performed using the proprietary AMP reagents, and the signal visualized through probe-specific horseradish-peroxidase-based detection by signal amplification with Opal 570 dyes (Perkin Elmer, 1:1500). Sections were incubated with secondary antibodies at RT. Slides were then counterstained with DAPI and coverslipped with Prolong Gold Anti-fade (ThermoFisher, P36935) and kept at 4 °C until imaging. Images were taken with a Leica Stellaris 8 microscope equipped with a White Light Laser (WLL) capable of tuning to excitation laser lines at 1 nm width between 440 nm and 790 nm.

## Flow cytometry and spectral flow cytometry

Initial steps of the enzymatic dissociation of the tumors are the same as described above, except 0.5% collagenase D (Sigma, 11088858001) and DNase I (Roche, 11284932001) were used in place of papain. Single-cell suspensions were passed through 70-µm cell strainers, centrifuged, and resuspended in 30% Percoll (GE Healthcare, 17-0891-01) solution containing 10% FBS (Hyclone SH30396.03). Cells were separated by centrifugation at $800 \times g$ for 15 min at 4 °C. The supernatant was carefully removed to discard debris and lipids. The cells were then washed in cold PBS and resuspended in RBC lysis buffer (BioLegend, 420301) for 1 min at 37 °C. Cells were transferred to an Eppendorf tube and washed once with FACS buffer (DPBS with 0.5% BSA) and blocked with 100 µl of 2× blocking solution (2% FBS, 5% normal rat serum, 5% normal mouse serum, 5% normal rabbit serum, 10 µg/ml anti-FcR (BioLegend, 101319) and 0.2% NaN$_3$ in DPBS) on ice for 30 min. Cells were then stained with primary antibodies (Supplementary Table 3, 1:100 dilution, except noted otherwise) on ice for 30 min and washed with PBS. The cells were subsequently incubated in 100 µl viability dye (Zombie UV, BioLegend, 1:800) at room temperature for 20 min. The cells were washed and fixed with fixation buffer (eBioscience, 00-5123-43, 00-5223-56) for 30 min at 4 °C. Cells were washed and stained with the cocktail of antibodies (1:50) examined myeloid lineage are set aside in the fridge until loading to the cytometer. Cells stained for the lymphoid panel were then permeabilized with a permeabilization buffer (eBioscience, 00-8333-56) before the intracellular markers were stained. The cells were washed and stored in the fridge till analysis. Antibodies used in this study include are listed in Supplementary Table 3. All data were collected on a BD LSR II flow cytometer or Cytek Aurora spectral flow cytometer. Data were analyzed offline using FlowJo 10 software (Tree Star Inc., RRID:SCR_008520).

## Cell cycle analysis

Cell cycle analysis was performed using propidium iodide (PI) solution (BD, 556463), and examined by flow cytometry. Briefly, cells ($1 \times 10^5$) grown in monolayer in 96-well plate were cultured at 37 °C with 5% CO$_2$ in the presence of TNF-α (concentration as indicated in the figures, Peprotech, 315-01 A) for 48 h. The cells were dissociated with 0.05% Trypsin (ThermoFisher, 25300054) to generate single-cell

suspensions. Following washing twice with cold PBS, the cells were fixed with 5 ml 70% cold ethanol for 30 min at 4 °C. Fixed cells were washed with PBS, and stained with 50 μl of PI buffer containing 20 μg/ml PI (BD Biosciences, 556463), 0.1% Triton X-100, 0.4 mg/ml RNase (ThermoFisher, EN0531). Cells were kept at 37 °C for 30 min in the dark. 400 μl of PBS was added to the cell suspension before they were examined on a flow cytometer (BD, Canto II, operated with Diva software). Cell cycles were determined using FlowJo plugins (FlowJo, BD Biosciences).

### TUNEL assay

Terminal deoxynucleotidyl transferase-mediated dUTP Nick End Labelling (TUNEL) assays were used for quantification of apoptotic cells following TNF-α treatment. Cells ($1 \times 10^5$) grown in a monolayer in 96-well plate were cultured for 48 h at 37 °C with 5% $CO_2$ in the presence of TNF-α (10 ng/ml, Peprotech, 315-01 A). The cells were fixed with 4% paraformaldehyde (Electron Microscopy Sciences, 15713-5) for 20 min at RT, followed by washing with PBS, permeabilization with 0.25% Triton X-100, and blocking with 3% bovine serum albumin (Fisher) in PBS. Apoptotic cells characterized by DNA fragmentation were detected using an Invitrogen Click-iT Plus TUNEL Assy for in situ Apoptosis Detection kit (Fisher C10617) following the manufacturer's instructions. The nuclei were counterstained with 40,6-diamidino-2-phenylindole (DAPI, Sigma). Two slides from each culturing condition were examined, four different field of view were acquired for each slide. Fluorescence and Differential Interference Contrast (DIC) images were taken on a Leica SP8 confocal microscope. Images were analyzed and quantified with FIJI with TUNEL$^+$ cells appear to be bright green.

### Glycolysis analysis

Cellular Glycolytic activity was assessed using the Glycolysis Assay kit (Abcam, ab197244) following the manufacturer's instructions. Cells ($6 \times 10^4$) grown in monolayer in 96-well plate were cultured for 24 h at 37 °C with 5% $CO_2$ in the presence of TNF-α (10 ng/ml, Peprotech, 315-01 A) at either normoxia (18% $O_2$) or hypoxia (1% $O_2$) conditions. Three hours prior to performing the Glycolysis assay measurement, $CO_2$ was purged by incubating cells in a $CO_2$-free incubator at 37 °C with 95% humidity. The cells were washed twice with Respiration buffer warmed to 37 °C. In all, 90 μl of the same buffer was then added to all wells containing cells and two other wells that were designated positive and blank controls. Overall, 10 μl Glycolysis Assay Reagent was then added to each well and positive control well. The prepared plate was immediately inserted into a Cytation 5 (Agilent) fluorescence plate reader pre-set to the measurement temperature at 37 °C. Glycolysis assay signal was measured at 1.5 min intervals for 120 min using excitation and emission wavelengths of Ex/Em = $360/610 \pm 20$ nm, respectively. The optimal delay time was set at 100 μs, and the integration time is set at 30 μs.

### Cell proliferation/viability assay

Cell proliferation/viability was examined using CellTiter 96 AQ$_{ueous}$ One Solution Cell Proliferation Assay (MTS) kit (Promega, G3580) following the manufacturer's instructions. Briefly, Cells ($1 \times 10^5$) grown in monolayer in a 96-well plate were cultured for 48 or 72 h at 37 °C with 5% $CO_2$ in the presence of TNF-α (10 ng/ml, Peprotech, 315-01 A). 20 μl of AQ$_{ueous}$ One Solution reagent was added to each well and the Cells were returned to the incubator for 2 h. Optic absorbance at 490 nm was then recorded using a Cytation 5 (Agilent) 96-well plate reader.

### scRNA-seq and data analysis

Single-cell suspensions of the tumors were obtained by papain dissociation as described above. The viability of single cells was assessed using Trypan Blue staining, and debris-free suspensions of >80% viability were deemed suitable for single-cell RNA Sequencing (scRNA-seq). Samples with lower viability were run with caution.

Single-cell RNA Seq was performed on these samples using the Chromium platform (10X Genomics) with the 3' gene expression (3' GEX) V3 kit, using an input of ~10,000 cells. Briefly, Gel-Bead in Emulsions (GEMs) were generated on the sample chip in the Chromium controller. Barcoded cDNA was extracted from the GEMs by Post-GEM RT-cleanup and amplified for 12 cycles. Amplified cDNA was fragmented and subjected to end-repair, poly A-tailing, adapter ligation, and 10X-specific sample indexing following the manufacturer's protocol. Libraries were quantified using Bioanalyzer (Agilent) and QuBit (ThermoFisher) analyses and were sequenced in paired-end mode on a NovaSeq instrument (Illumina) targeting a depth of 50,000–100,000 reads per cell.

Raw fastq files were aligned to mouse genome reference mm10 customized to include the Rfp sequence, using CellRanger v5.0.0 (10X Genomics). Count matrices filtered by CellRanger algorithm were further filtered by discarding cells with either <200 genes, <1000 UMI (unique molecular identifier), or >25% mitochondrial genes expressed. Data was processed and analyzed using R package Seurat v4.0.5. Normalization was performed using NormalizeData function with normalization.method = "LogNormalize". Dimensionality reduction was computed on the top 2000 variable features using FindVariableFeatures, ScaleData and RunPCA functions. UMAPs were generated using the top 15 PCs. For subclustering the immune compartment, we used R package Harmony (v1.0) to mitigate for batch effects driven by technical variation between replicates. De novo clustering using the Louvain algorithm was applied at different resolutions (0.2; 0.8; 2; 5; 8) on the SNN graph space. For high-level annotation, cell classes were identified in an iterative and semi-supervised fashion by assigning de novo discovered clusters to cell classes based on expression of known marker genes that define each cluster. Annotation of cell subtypes at a lower-level was performed in a similar manner as for the high-level and further aided by de novo marker discovery using the Seurat FindMarkers function and Wilcoxon rank-sum test for differential expression analysis. To identify doublet-enriched clusters, we looked for clusters of cells displaying expression of canonical markers for two or more different cell types and higher number of genes/UMI; such clusters were removed from further analysis.

In this study, we inferred cell–cell interactions using CellPhoneDB (version 2.1.7)[60], which utilizes a curated, publicly available repository of ligand-receptor interactions. We calculated the ligand-receptor interaction for each sample separately through the permutation method implemented by CellPhoneDB and visualized the differential enrichments using a dotplot, where the colors represented the differential enrichment between *WT; Ntv-a* and *qMCP$^{-/-}$; Ntv-a* samples and the size of the dot corresponded to the Fischer exact test *P* values comparing the difference in significant interactions between *WT; Ntv-a* and *qMCP$^{-/-}$; Ntv-a* samples.

Identification of modules of co-expressed genes was carried out using the R package scWGCNA (v0.2.11) (https://github.com/smorabit/scWGCNA) by first computing meta cells of 100 neighboring cells (k = 100) using the function construct_metacells. To identify modules, function blockwiseConsensusModules was called with the following parameters: softPower=12, deepSplit=3, mergeCutHeight = 0.25. Only the top 2000 variable genes were used. Module scores, representing a normalized average expression of all genes in the WGCNA module, were computed using Seurat function AddModuleScore. Pathway enrichment analysis of gene modules identified using WGCNA was carried out using R package clusterProfiler (v4.2.0)[101], using the GSEA MSigDB HALLMARK gene set annotation for mouse genes (downloaded at https://bioinf.wehi.edu.au/MSigDB/ on 09/08/2022).

To generate the pie charts shown in Supplementary Figs. 3A and 17B, we tally the total counts of the molecules-of-interest and traced their source to either malignant cells or non-malignant cells. The total counts obtained from the tumor compartment was then divided by the number of tumor cells expressing these molecules. Similarly, this ratio

was also obtained from the non-malignant compartment. These two ratios were subsequently used to create the pie charts.

## Statistical analyses

Graphs were created using GraphPad Prism 9 (GraphPad Software Inc.) or R. Variables from two experimental groups were analyzed using unpaired or paired parametric two-tailed $t$ tests as appropriate, assuming equal standard deviations. One-way ANOVA was used to compare variables from more than two groups. Kaplan–Meier survival analysis was performed using the Log-rank (Mantel–Cox) test and Gehan–Breslow-Wilcoxon test. Further details are included in the figure legends. Power analysis was performed based on prior experimental results obtained in the lab, with consideration of 10% attrition rate due to unexpected events such as spontaneous sarcoma, dermatitis or fight wound. $*P < 0.05$; $**P < 0.01$; $***P < 0.001$; $****P < 0.0001$; (ns) not significant. Final figures were assembled in Creative Cloud Photoshop (Adobe, RRID:SCR_014199).

## Reporting summary

Further information on research design is available in the Nature Portfolio Reporting Summary linked to this article.

## Data availability

ScRNA-Seq data were deposited at GEO with accession number GSE203154. A publicly available raw mRNA expression data, age, sex and GBM survival outcomes were downloaded from selected study Glioblastoma Multiforme (TCGA, Firehose Legacy cBioPortal [https://www.cbioportal.org]. The remaining data are available within the Article, Supplementary Information or Source Data file. Source data are provided with this paper.

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

## Acknowledgements

We would like to acknowledge the Mouse Transgenic and Gene Targeting Core, Flow Cytometry Core and the Integrated Cellular Imaging Cores at Emory University for their services. We would also like to acknowledge the Mount Sinai Dean's Flow Cytometry Core and Genomics Core for scRNA-seq services. The Tisch Cancer Institute and related research facilities are supported by P30 CA196521. We thank Dr. Jia Liu at the Epigenetics Core of the City University of New York for performing the RNAScope assay. We thank Erin Moshier and Mayuri Jain at the Biostatistics Shared Resource Facility at Mount Sinai for Cox regression analysis. We extend our thanks to Mr. David R. Schumick for generating illustrations and Dr. Christopher Nelson for scientific editing. Supplementary Figs. 8, 21B, and 25A were created with BioRender.com. This work was supported by NIH/NINDS R01 NS100864 and start-up funds to DH from the Departments of Oncological Sciences and Neurosurgery, Icahn School of Medicine, Mount Sinai. N.M.T. and D.H. were supported by R01NS106229. A.L. was supported by Damon Runyon-Rachleff Innovation Award (DR52-18) and R37 Merit Award (R37CA230636), and Icahn School of Medicine at Mount Sinai. E.E.B.G. was supported by NIH/NIAID R01AI123126.

## Author contributions

Concept and design: Z.C. and D.H. Development of Methodology: Z.C., A.L., E.E.B.G., A.M.T., and D.H. Acquisition of the data (provided animals, acquired, and managed patients, provided facilities, etc.): Z.C., G.P., D.J.E., K.E.L., J.L.R., M.P.V., T.J., A.A., W.T., A.K., N.M.T., D.H.G., E.E.B.G., S.A.L., and D.H. Analysis and interpretation of the data (e.g., statistical analysis, biostatistics, computational analysis): Z.C., N.S., G. G.P., B.G., D.J.E., J.L.R., E.E.B.G., A.M.T., and D.H. Writing, review, and/or revision of the manuscript: D.H. created the original draft. All authors participated in the review and editing. Administrative, technical, or material support (i.e., reporting or organizing the data, constructing databases): Z.C., N.S., and B.G. Study supervision: Z.C. and D.H.

## Competing interests

The authors declare no competing interests.

## Additional information

[1]Department of Oncological Sciences, The Tisch Cancer Institute, Icahn School of Medicine at Mount Sinai, New York, NY 10029, USA. [2]Department of Pediatrics, Aflac Cancer and Blood Disorders Center, Children's Healthcare of Atlanta and Winship Cancer Institute, Emory University School of Medicine, Atlanta, GA 30322, USA. [3]Winship Cancer Institute, Emory University School of Medicine, Atlanta, GA 30322, USA. [4]Department of Genetics and Genomic Sciences, Icahn School of Medicine at Mount Sinai, New York, NY 10029, USA. [5]Department of Medicine, Lowance Center for Human Immunology and Emory Vaccine Center, Emory University School of Medicine, Atlanta, GA 30322, USA. [6]Liver Cancer Program, Division of Liver Diseases, Department of Medicine, Tisch Cancer Institute, Icahn School of Medicine at Mount Sinai, New York, NY 10029, USA. [7]The Precision Immunology Institute, Icahn School of Medicine at Mount Sinai, New York, NY 10029, USA. [8]Graduate School of Biomedical Sciences at Icahn School of Medicine at Mount Sinai, New York, NY 10029, USA. [9]Emory University Department of Microbiology and Immunology, Emory Vaccine Center, Atlanta, GA 30322, USA. [10]Department of Pathology and Laboratory Medicine, Icahn School of Medicine at Mount Sinai, New York, NY 10029, USA. [11]Department of Neurology, Washington University School of Medicine, St. Louis, MO 63110, USA. [12]Precision Immunology Institute, Icahn School of Medicine at Mount Sinai, New York, NY 10029, USA. [13]Department of Neurosurgery, Icahn School of Medicine at Mount Sinai, New York, NY 10029, USA. [14]These authors contributed equally: Nishant Soni, Gonzalo Pinero. ✉e-mail: zhihong.chen@mssm.edu; dolores.hambardzumyan@mssm.edu

