## [Peer Review File · Nature Communications]

Monocyte depletion enhances neutrophil influx and Proneural to Mesenchymal transition in GlioblastomaEditorial Note: This manuscript has been previously reviewed at another journal that is not operating a transparent peer review scheme. This document only contains reviewer comments and rebuttal letters for versions considered at *Nature Communications*.

REVIEWERS' COMMENTS

Reviewer #1 (Remarks to the Author):

The authors have now addressed all critiques raised in the initial submission and the manuscript is in very good shape.

Reviewer #3 (Remarks to the Author):

The authors have been attentive to some issues raised and the manuscript has been strengthened by the addition of new data. With that said, there are few key comments that were not fully addressed. For examples, I totally agree with the #1 reviewer that the glioma model used in this study may not be appropriate. Based on my original comments, the authors provided additional data in Nf1-silenced tumor model. However, they did not validate their key findings by developing combinational therapies to inhibit both monocytes and neutrophils simultaneously in this model.

Question #5: the authors provided neutrophil quantification using flow cytometry (Fig. 1L). There is no difference found between two groups, which is inconsistent with the conclusion of the paper, and they even did not cite this finding in the manuscript. I am not understand for why the authors decided to perform IBA1 IHC staining here (It is not helpful for supporting the conclusion of the manuscript or addressing the concern). If the authors wanted to distinguish BMDM and microglia, they may want to use co-staining for IBA1 and CD49D.

Question #11: The authors did not answer the question regarding whether the PN-MES transition is triggered by neutrophil infiltration or by the direct effect of MCPs on glioma cells directly.

Reviewer #4 (Remarks to the Author):

Zhihong Chen et al. generated a MCPs-deficient mouse model of glioblastoma and hepatocellular carcinoma via CRISPR/Cas9 genome editing, and found monocyte and neutrophil drive glioblastoma progression. They showed neutrophil cells can promote proneural-to-mesenchymal transition and increased hypoxia in PDGFRB-driven glioblastoma via TNF- α . Inhibiting or decreasing neutrophil can improve the survival in PN and MES phenotype glioblastoma.

Overall, this is one interesting manuscript, but my comments are below,

There are several single cell studies showing the cell plasticity and tumor heterogeneity for glioblastoma, for example Cyril Neftel, *Cell*, 2019, Lin Wang, *Cancer Discovery*, 2019 and Laura M. Richards, *Nature Cancer*, 2021. These papers show that PN and MES phenotype are the major variance in glioblastoma. When the author discussed the PN and MES, these important papers should be included for discussion. The author claimed that neutrophil cells can promote proneural-to-mesenchymal transition and increased hypoxia in PDGFRB-driven glioblastoma via TNF- α . The shift of proneural-to-mesenchymal transition often occurs in glioblastoma after treatment (e.g. RT, TMZ). Recently, there's one *Nature Cancer* paper profiled paired primary and recurrent glioblastoma via single cell multi-omics (<https://www.nature.com/articles/s43018-022-00475-x>). The author should use recurrent glioblastoma data to validate their conclusions, for example, does the percentage/signature of neutrophil cell increase in recurrent glioblastoma or not? Are there any significant differences for neutrophil cell between primary and recurrent?

Reviewer #3 (Remarks to the Author)

Q1. The authors have been attentive to some issues raised and the manuscript has been strengthened by the addition of new data. With that said, there are few key comments that were not fully addressed. For examples, I totally agree with the #1 reviewer that the glioma model used in this study may not be appropriate. Based on my original comments, the authors provided additional data in Nf1-silenced tumor model. **However, they did not validate their key findings by developing combinational therapies to inhibit both monocytes and neutrophils simultaneously in this model.**

New Supplementary Figure 31. Genetic loss of *Cxcl1* in monocyte abolished tumors results in decreased neutrophil infiltration and extended survival of *Nf1*-silenced tumor-bearing mice.

A1. We provided data that abolishing monocyte recruitment does not affect neutrophil influx and survival of *Nf1*-silenced GBM-bearing mice (**Fig. S30**). Decreasing neutrophil influx did not affect monocyte recruitment and resulted in extended survival time of tumor-bearing mice. We hypothesized that the combined targeting of both populations would not seem synergistic in *Nf1*-silenced tumors.

We performed genetic experiments, and results from abolishing monocytes and decreasing neutrophil recruitment led to extended survival time of tumor-bearing mice compared to monocyte-abolished tumors (We provide a new supplementary figure 31) and a similar extent to only neutrophil targeting.

Q2. a) Question #5: the authors provided neutrophil quantification using flow cytometry (Fig. 1L). There is no difference found between two groups, which is inconsistent with the conclusion of the paper, and they even did not cite this finding in the manuscript.

b) I am not understand for why the authors decided to perform IBA1 IHC staining here (It is not helpful for supporting the conclusion of the manuscript or addressing the concern). If the authors wanted to distinguish BMDM and microglia, they may want to use co-staining for IBA1 and CD49D.

A2. a) In response to reviewer 3 Q5, which was also raised by Reviewer 1, we referred to responses we provided to reviewer 1; we apologize we should have answered for Reviewer 3 below section Q5. The baseline expression levels for *Ccl2*, *Ccl7*, *Ccl8*, and *Ccl12* in various tumor cell and microenvironmental cell populations from scRNA-seq data from WT and MCP-deficient tumors were all provided in Supplemental Fig. S7A (original version, now Fig. S10). To provide better clarity, we have now combined malignant and non-malignant cell types from the tumor microenvironment into two groups as a pie chart in a new Supplemental Fig. S3A.

Figure S3. (A) Pie chart of quantification of CCL expression between non-malignant and malignant cells based on scRNA-seq data from *PDGFB*-driven GBM in *WT;Ntv-a* mice.

The results clearly illustrate that although larger percent of Ccl2, Ccl7, Ccl8, and Ccl12 are expressed by non-malignant cells of TME, still, tumor cells express a significant fraction. Deletion of MCPs from the stromal compartment only was achieved by transplanting WT tumors into WT and MCP KO recipients as presented in Fig. 1J-K. In these mice, MCP levels were decreased (due to the tumor cells being WT for MCPs), which resulted in a decrease in monocyte recruitment and extended survival time of tumor-bearing mice in the MCP KO background with no changes in neutrophil recruitment, which we discussed in the manuscript. Since MCPs are also produced by tumor cells, to avoid potential compensatory increase of their expression by tumor cells when the MCPs are deleted from the TME, and to further decrease their production to achieve maximum target inhibition and abolish monocyte infiltration (Fig. 2A), we generated tumors in mice that lack MCPs both from tumor cells and the TME. While we achieved our goal of abolishing monocyte influx, we also observed compensatory neutrophil recruitment.

Q2. b) am not understand for why the authors decided to perform IBA1 IHC staining here (It is not helpful for supporting the conclusion of the manuscript or addressing the concern). If the authors wanted to distinguish BMDM and microglia, they may want to use co-staining for IBA1 and CD49D.

A2. b) *This experiment was performed in response to reviewer 2 Q6b. This was to compare results with IBA1 staining in tumors that lack MCPs only in TME versus tumors that lack MCPs from both TME and tumor cells, where we documented a significant reduction of IBA1 positive area (Figure 4F, we also used P2RY12 staining to distinguish microglia from BMDM). FACS data distinguishing BMDM and microglia in this transplant model were provided in Figure 1L.*

Q3. Question #11: The authors did not answer the question regarding whether the PN-MES transition is triggered by neutrophil infiltration or by the direct effect of MCPs on glioma cells directly.

A3. *Does this contribute to PN-MES transition? To address this question, we used series of newly generated early passage primary WT-GSC and qMCP^{-/-}-GSC cultures (n = 3 per each genotype). These cultures were maintained in*

Figure 5. (E) Quantitative real-time PCR examine the expression of signature MES genes in primary PDGFB-driven GBM tumor cell cultures derived from *WT;Ntv-a* (WT-GSC) and *qMCP^{-/-};Ntv-a* (*qMCP^{-/-}* GSC) mice (n = 3 independent primary cultures for each genotype).

*serum-free medium as spheres, and experiments were performed on cultures up to 5-6 passages. We evaluated expression of several MES signature genes in these cultures but did not observe differences in the expression of MES signature genes between WT and *qMCP^{-/-}*-GSC cultures (new Fig. 5E).*

Figure 5. (F) Quantitative real-time PCR examining the expression of signature MES genes in primary PDGFB-driven WT GBM tumor cell cultures treated with 10 ng/mL of TNF- α for 48 hrs.

*We then stimulated WT and *qMCP^{-/-}*-GSC cells with recombinant TNF α (0, 10, and 50 ng/ml) for a period of 48 hrs and the expression of several genes enriched in MES signature analyzed. There was increased expression of MES genes in both WT (new Fig. 5F) and *qMCP^{-/-}*-GSC cultures (new Fig. S24) in response to TNF α stimulation. We found that all of these genes are increased in after TNF α stimulation, providing direct evidence that TNF α , but not loss of qMCPs, contributes to PN->MES transition.*

Figure S24. TNF- α stimulation of qMCP-deficient GSC cultures induces expression of MES signature genes. Quantitative real-time PCR for expression of MES signature genes in qMCP-deficient primary GSC cultures treated with 10 ng/mL of TNF- α for 48 hrs. Student's paired *t*-test compared to non-treated cells. * $p < 0.05$, ** $p < 0.001$, *** $p < 0.0001$.

Reviewer # 4

Q1. There are several single cell studies showing the cell plasticity and tumor heterogeneity for glioblastoma, for example Cyril Neftel, Cell, 2019, Lin Wang, Cancer Discovery, 2019 and Laura M. Richards, Nature Cancer, 2021. These papers show that PN and MES phenotype are the major variance in glioblastoma. When the author discussed the PN and MES, these important papers should be included for discussion.

A1. We have discussed Neftel and colleagues' manuscript that described a model with four potentially plastic cellular states in human GBM: AC-like, MES-like NPC- and OPC-like. We now will add to the introduction that others using a combination of high-dimensional technologies, including scRNA and snRNA-sequencing, have documented that GBM contains hierarchies of MES and PN GSCs and their more differentiated progeny both in primary¹ and recurrent tumors². Recurrent GBM showed an increased number of cells with MES signature compared to primary matched samples, suggesting that SOC drives PN->MES transition². Others explained GBM heterogeneity using the GSC model and showed that they exist along a major transcriptional gradient between two cellular states, developmental and Injury response programs³.

Q2. The author claimed that neutrophil cells can promote proneural-to-mesenchymal transition and increased hypoxia in PDGFRB-driven glioblastoma via TNF- α .

A2. We provided direct evidence to back up the claim that TNF- α can induce PN->MES transition in our PDGFRB-driven GBM model and further validated the transition using *in vivo* models. Similar results *in vitro* were documented in human PN GBM GSC cultures showing that TNF- α via activation of NF- κ B in tumor cells can drive PN->MES transition, which we discussed in the manuscript⁴. In addition, our results identified and validated neutrophils as the source of TNF- α .

Q3. The shift of proneural-to-mesenchymal transition often occurs in glioblastoma after treatment (e.g. RT, TMZ). Recently, there's one Nature Cancer paper profiled paired primary and recurrent glioblastoma via single cell multi-omics (<https://www.nature.com/articles/s43018-022-00475-x>). The author should use recurrent glioblastoma data to validate their conclusions, for example, does the percentage/signature of neutrophil cell increase in recurrent glioblastoma or not? Are there any significant differences for neutrophil cell between primary and recurrent?

A3. Within the scope of one manuscript, we focused on myeloid cells from the blood and how their interplay results in PN->MES transition. In our manuscript, (1) we performed a comprehensive analysis of

chemokines involved in neutrophil recruitment (CXCL1, CXCL2, CXCL3, and CXCL5), (2) demonstrated at both the RNA and protein level that these chemokines are enriched not only in MES, but also in NF1^{del/mut} tumors (which cluster in MES signature group), relative to PN and PDGFRA-amplified human GBM, (3) performed immune specific RNA profiling of our primary GBM patient collection that we have previously published subtypes⁵, and (4) defined a neutrophil score on the same sections stained for elastase, which were employed for proteomic analysis to validate chemokines at the protein level.

It has been demonstrated by others and us that the PDGFB-driven model we used in this manuscript undergoes PN->MES transition in response to RT⁶ and anti-VEGFA treatment⁷. Evaluating neutrophil response in primary versus recurrent tumors is beyond the scope of this manuscript; nonetheless it is an exciting question for future studies, that would determine whether PN->MES transition mechanisms and drivers can be different depending on therapy used.

References

1. Wang, L., *et al.* The Phenotypes of Proliferating Glioblastoma Cells Reside on a Single Axis of Variation. *Cancer Discov* **9**, 1708-1719 (2019).
2. Wang, L., *et al.* A single-cell atlas of glioblastoma evolution under therapy reveals cell-intrinsic and cell-extrinsic therapeutic targets. *Nat Cancer* **3**, 1534-1552 (2022).
3. Richards, L.M., *et al.* Gradient of Developmental and Injury Response transcriptional states defines functional vulnerabilities underpinning glioblastoma heterogeneity. *Nat Cancer* **2**, 157-173 (2021).
4. Bhat, K.P.L., *et al.* Mesenchymal differentiation mediated by NF-kappaB promotes radiation resistance in glioblastoma. *Cancer Cell* **24**, 331-346 (2013).
5. Kaffes, I., *et al.* Human Mesenchymal glioblastomas are characterized by an increased immune cell presence compared to Proneural and Classical tumors. *Oncoimmunology* **8**, e1655360 (2019).
6. Halliday, J., *et al.* In vivo radiation response of proneural glioma characterized by protective p53 transcriptional program and proneural-mesenchymal shift. *Proc Natl Acad Sci U S A* **111**, 5248-5253 (2014).
7. Pitter, K.L., *et al.* Corticosteroids compromise survival in glioblastoma. *Brain* **139**, 1458-1471 (2016).